# Ecological trade-offs drive phenotypic and genetic differentiation of *Arabidopsis thaliana* in Europe

Cristina C. Bastias [1,2,6] ✉, Aurélien Estarague [1,3,6], Denis Vile [3], Elza Gaignon [1], Cheng-Ruei Lee [4], Moises Exposito-Alonso [5], Cyrille Violle [1,7] & François Vasseur [1,7]

Plant diversity is shaped by trade-offs between traits related to competitive ability, propagule dispersal, and stress resistance. However, we still lack a clear understanding of how these trade-offs influence species distribution and population dynamics. In *Arabidopsis thaliana*, recent genetic analyses revealed a group of cosmopolitan genotypes that successfully recolonized Europe from its center after the last glaciation, excluding older (relict) lineages from the distribution except for their north and south margins. Here, we tested the hypothesis that cosmopolitans expanded due to higher colonization ability, while relicts persisted at the margins due to higher tolerance to competition and/or stress. We compared the phenotypic and genetic differentiation between 71 European genotypes originating from the center, and the south and north margins. We showed that a trade-off between plant fecundity and seed mass shapes the differentiation of *A. thaliana* in Europe, suggesting that the success of the cosmopolitan groups could be explained by their high dispersal ability. However, at both north and south margins, we found evidence of selection for alleles conferring low dispersal but highly competitive and stress-resistance abilities. This study sheds light on the role of ecological trade-offs as evolutionary drivers of the distribution and dynamics of plant populations.

The range of a species often encompasses a wide diversity of habitats, each exerting distinct selection pressures and ultimately resulting in different phenotypes among populations[1,2]. In plants, phenotypic diversity is constrained by major trade-offs between traits related to contrasting functions, such as stress tolerance, competitive ability, and reproductive investment. These trade-offs collectively define alternative ecological strategies[3,4], which differ among populations according to geographical location and evolutionary history[5,6].

However, the role of ecological trade-offs in delineating species distribution and population dynamics remains unclear.

Because populations experience contrasted climatic conditions, phenotypic traits often correlate with major climatic gradients, such as latitude. For instance, leaf size has been shown to globally increase in low latitudes with hot and wet environments[7], while plant size tends to increase at high latitudes[8]. However, phenotypic differentiation can also follow a center-to-margin gradient, as observed in different

[1]CEFE, Univ Montpellier, CNRS, EPHE, IRD, Montpellier, France. [2]Área de Ecología, Facultad de Ciencias, Universidad de Córdoba, Campus de Rabanales, Córdoba, Spain. [3]LEPSE, Univ Montpellier, INRAE, Institut Agro Montpellier, Montpellier, France. [4]Institute of Ecology and Evolutionary Biology & Institute of Plant Biology, National Taiwan University, Taipei, Taiwan. [5]Department of Plant Biology, Carnegie Institution for Science, Stanford, CA, USA. [6]These authors contributed equally: Cristina C. Bastias, Aurélien Estarague. [9]These authors jointly supervised this work: Cyrille Violle, François Vasseur. ✉e-mail: crbasc@gmail.com

species[9–12]. This is explained by different ecological and evolutionary mechanisms. First, it has been suggested that populations located at the peripheries of a species' distribution will encounter the most challenging environmental conditions that the species can endure[13,14]. Peripheral populations are thus expected to display traits related to stress tolerance and resource conservation. In addition, population demography is expected to differ between the center and the margins of the distribution range. For instance, smaller population sizes at the range margins should increase the effect of genetic drift on phenotypic divergence[15–17]. Moreover, intraspecific competition shows reduced intensity and exerts smaller fitness effects at the range margins in contrast to the range center[18,19]. However, we still lack a comprehensive understanding of how traits related to stress response and competitive abilities vary across a species distribution range, notably between the center and the margins.

In plant species that release their seeds within a limited area and timeframe, competition among related seedlings can be particularly intense, as they compete for finite resources within a restricted space[20–22]. Tolerance and avoidance are two commonly adopted strategies by plants to cope with competition[23–25]. The tolerance strategy involves minimizing the negative impacts of competition from neighboring plants by producing large well-supplied seeds that increase the future competitive capacity of the seedlings[4,26]. In contrast, the avoidance strategy aims to reduce proximity to competitors through the production of many small seeds, which disperse more easily and limit their spatial aggregation[27,28]. Accordingly, the competition-colonization trade-off suggests a plant cannot excel simultaneously at being both a strong competitor and an efficient disperser[29]. Moreover, traits that influence competitive ability, such as plant growth and seed mass, are also expected to play a significant role in determining plant's responses to abiotic stresses, thereby creating a related trade-off known as the colonization-stress tolerance trade-off[30,31]. For instance, populations at the range margins, as opposed to those at the center, are expected to experience stronger selection for stress tolerance and survival, associated with a lower investment in reproduction. However, empirical evidence of the competition-colonization trade-off (as well as the colonization-stress tolerance trade-off) remains ambiguous and is mostly limited to large comparative studies across diverse plant species[32–35] (but see[36–38]).

In the genomic era, intraspecific analysis of genetic sequences serves as compelling evidence for understanding past evolutionary processes and providing hypotheses regarding population adaptation, demography, and range expansion. This approach has been successfully applied to the widely distributed species *Arabidopsis thaliana* to trace the dynamics of its European populations[39,40]. The research conducted by Lee and colleagues[40] suggested that approximately 20,000 years ago, while an ice cap covered the central and northern parts of Europe, ancestral populations of *A. thaliana*, referred to as "relicts", were confined to southern Europe. As the ice cap began to melt and the geographical barrier dissolved, populations from these relict groups expanded northwards, reaching what is today northern Sweden and Russia. In a second phase, at 10 ka (or maybe earlier, as suggested by a recent study[41]), a group originating from central Europe (Balkans or Caucasus[41]) emerged as the dominant one, surpassing all the other genetic groups. It has been estimated that around 95% of the present-day European genotypes, called "cosmopolitan" hereafter, are descended from this second colonization wave. Genetic analyses of the 1001 genomes dataset[39] have unveiled that the present representatives of the relict group are currently confined to the Mediterranean basin, with a notable concentration in Spain[42,43]. In contrast, cosmopolitan genotypes are dispersed throughout Europe, organized into eight distinct genetic clusters[39].

The distribution and phenotypic diversity of cosmopolitan groups across Europe is constrained by climate-associated factors associated with latitude[44]. The most suitable habitats are found in central and western Europe. In contrast, low suitability is observed in northern and southern Europe, which is explained by winter cold and summer heat, respectively. Accordingly, phenotypic traits differ between south, center, and north genotypes[8,9,45–47]. For instance, there is strong evidence for a latitudinal gradient of flowering time and resource-use traits[46,48–51]. Genotypes from the Mediterranean basin, which have evolved in peculiar climatic conditions and under partial isolation (*e.g.*, Iberian Peninsula separated by the Pyrenees mountains, Italian Peninsula separated by the Alps), display early flowering, fast growth, and resource-acquisitive traits. By contrast, populations from the Scandinavian Peninsula (which also evolved under specific conditions and partial isolation) display late flowering, slow growth, and resource-conservative traits. However, whether this latitudinal gradient observed in cosmopolitan genotypes extends to relict genotypes remains unclear. For instance, most Iberian relics show late flowering[39] in contrast to Iberian cosmopolitans[52]. Interestingly, a genetic cluster of cosmopolitan genotypes from North Sweden, *i.e.* situated at the opposite end of the range compared to current relicts, showed high genetic similarities to Iberian relict genotypes[40,53]. This suggests that phenotypic variation in *A. thaliana* could follow a center-to-margin gradient, related to the genetic divergence from the cosmopolitan origin (central Europe) towards relict refugia at opposite latitudes (north and south Europe). Accordingly, there is genomic evidence for hybridization between relicts and cosmopolitans in both north and south Europe[53–56], although it is still unclear how genetic exchanges between the two groups have contributed to phenotypic evolution and adaptation at the margins.

Although the initial colonization of *Arabidopsis thaliana* in Europe by relicts can be traced back to the gradual melting of the ice cap, the success of the cosmopolitan group's expansion across Europe remains enigmatic, as does the preservation of relictual genetic variations solely at the peripheries of its range. In the present study, we investigated three questions: (*i*) do traits differentiate between *A. thaliana* populations according to a center-to-margin gradient, with northern lines displaying similar trait variations as their genetic relatives at the opposite southern margin? (*ii*) does trait variation among geographical and genetic groups relate to major ecological trade-offs between stress resistance, competition tolerance, and dispersal ability? and (*iii*) how does trait variation account for the maintenance of relictual, genetic variation at the opposite margins of the distribution range? To address these questions, we conducted experiments using 71 natural genotypes from three geographical areas (center, south, and north) within the European range of *A. thaliana*. We investigated phenotypic and genetic correlations between traits and explored whether phenotypic differentiation toward the range margins could be explained by adaptive introgressions following hybridization between cosmopolitans and relicts.

In this work, we show evidence that a trade-off between competition- and colonization-related traits modulates the demography of European populations of *Arabidopsis thaliana*. In cosmopolitan populations, increased fecundity has facilitated their successful colonization after the last glaciation in Europe. Conversely, marginal populations have managed to survive in the most southern and northern of Europe by beneficiating from alleles with ancestral origins, which confer advantages in stress resistance and competitive ability. Accordingly, our study suggests the presence of adaptive introgressions following hybridization between cosmopolitans and relicts at the range margins.

## Results

### A center-to-margins differentiation of *A. thaliana* populations across the European distribution range

To study the diversity and distribution of plant traits across Europe, we used 71 European genotypes of *A. thaliana*, including seven genotypes previously described as relicts[39]. Trait measurements on plants grown

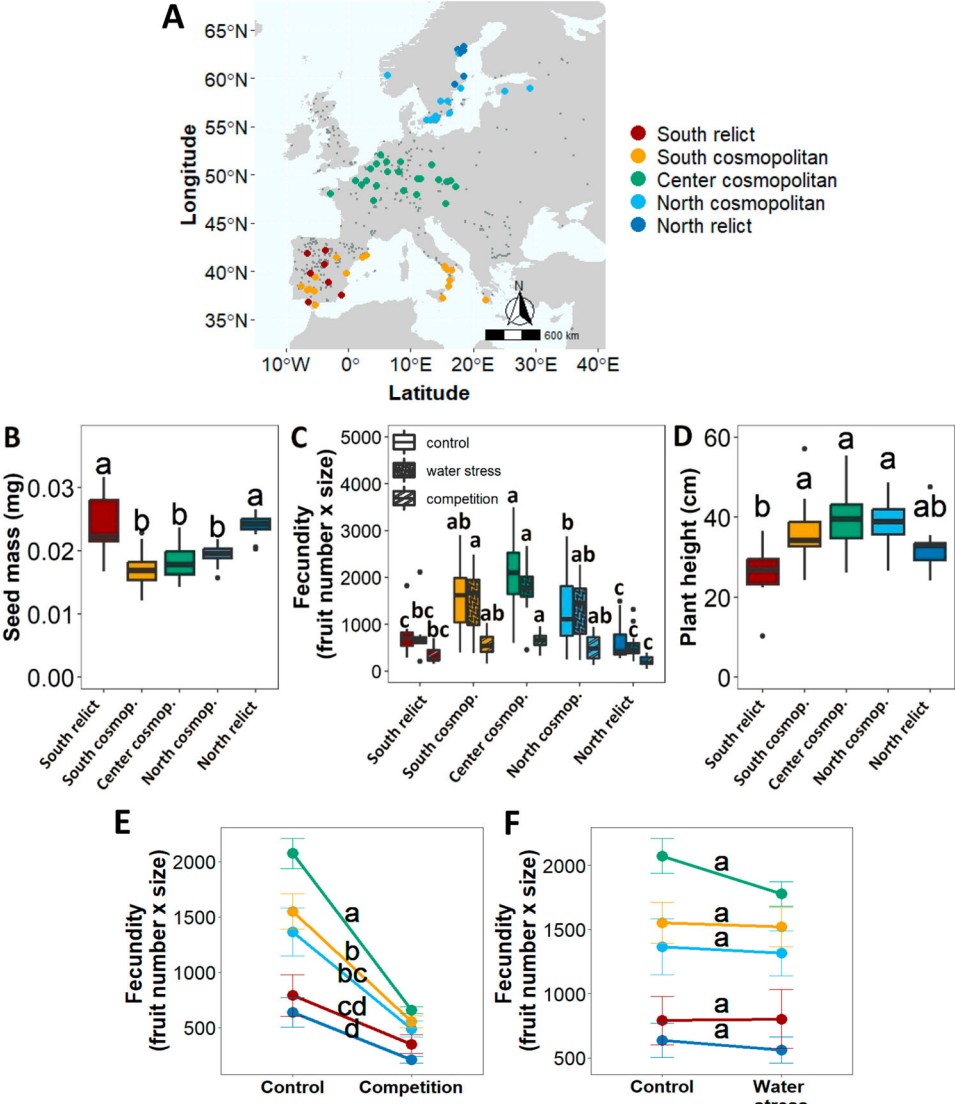

**Fig. 1 | Genetic and phenotypic differentiation among A. thaliana biogeographical groups.** The 71 genotypes ($n = 11$ for North relict, $n = 13$ for North cosmopolitan, $n = 23$ for Center cosmopolitan, $n = 17$ for South cosmopolitan, and $n = 7$ for South relict group) selected for experimentation in this study are depicted according to geographical and genetic clustering (**A**). Phenotypic differences in seed mass (**B**), fecundity under control, intraspecific competition and water stress conditions (**C**), and plant height (**D**) among biogeographical groups. Plot boxes (**B**–**D**) show the minimum, first quartile, median, third quartile, and maximum value. Reaction norms of fecundity in response to intraspecific competition (**E**) and water stress (**F**) among biogeographical groups. Data in **E**, **F** panels are presented as mean values +/− SEM. Different letters indicate significant differences in the mean trait between groups after Tukey's HSD test.

under non-stressing conditions (no competition and well-watered, *i.e.* 'control' hereafter; Supplementary Fig. 1) revealed non-linear relationships with latitude (Supplementary Fig. 2) for fecundity (number of seeds produced), seed mass, and plant height. This suggested that traits related to dispersal and competition abilities displayed a center-to-margin differentiation. Accordingly, measurements of the differences in the genotypic mean fecundity between individuals grown in intraspecific competition or under water stress and those in control ('fecundity response to competition' and 'fecundity response to water stress', respectively, hereafter) showed similar latitudinal trends (Supplementary Fig. 2). Overall, genotypes from intermediate latitudes were taller, produced more but smaller seeds, and were more impacted by competition and water stress, than genotypes from low and high latitude.

To further investigate the distribution of plant traits, we grouped genotypes according to their geographical and genetic origins. We first defined three geographical areas representative of center, south, and

north Europe, which are separated by natural barriers that partially isolate plant populations. We used the latitude threshold of 45°, which corresponds to the Pyrenees and Alps mountains, to delineate the south area, and the latitude threshold of 55° that separates the Scandinavian Peninsula[39] (Fig. 1A). We then employed ADMIXTURE to explore the genetic structure of the sample. This analysis revealed that the 71 genotypes can be classified into two genetic clusters (displaying the lowest cross-validation error when k ranges from 1 to 9 clusters; Supplementary Fig. 3). The first genetic cluster ($n = 53$ genotypes) included all genotypes from the central Europe area and a portion of those located at the south and north regions. All these genotypes belonged to one of the eight cosmopolitan groups identified in the 1001 genomes[39] (Supplementary Table 1), and are henceforth referred to as the cosmopolitan group. The second genetic cluster ($n = 18$) comprised all relict genotypes from the Iberian Peninsula, as well as all genotypes from North Sweden. This second cluster confirmed the high genetic relatedness between relicts and North Sweden lines previously

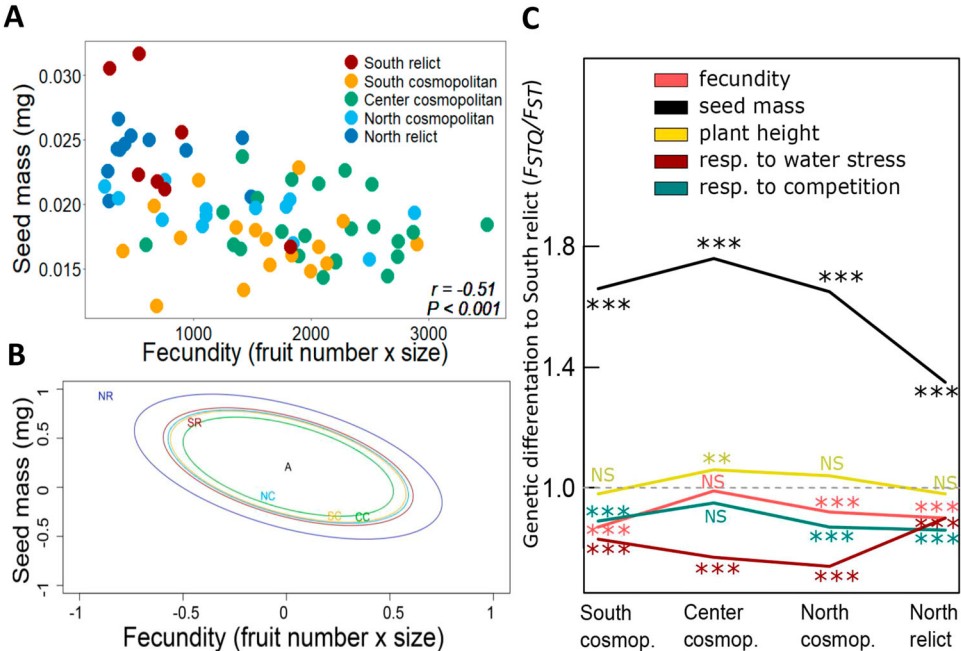

**Fig. 2 | Competition vs. colonization trade-off and genetic differentiation among biogeographical groups of A. thaliana.** **A** Relationship between seed mass, estimated from seeds (*n* = 10-30 air-dried seeds) produced by 4-5 individuals per *A. thaliana* genotype, and fecundity measured as the total number of fruits per genotype multiplied by the average fruit length (*n* = 8 individual plants per genotype). Each point represents the mean trait value of one genotype (In total *n* = 71 genotypes; *n* = 11 for North relict, *n* = 13 for North cosmopolitan, *n* = 23 for Center cosmopolitan, *n* = 17 for South cosmopolitan, and *n* = 7 for South relict group). Pearson's correlation coefficient (*r*) and statistical significance (*P*) are shown. **B** $Q_{ST}$ $F_{ST}$ test in a multivariate framework using the method of Ovaskainen and colleagues[59] and testing seed mass and fecundity trait divergence expected by genetic drift only among biogeographical groups. Solid ellipses show the phenotypic divergence expected by neutral processes only to the estimated ancestral

state (noted by the A label). Labels represent the observed phenotypic divergence of each biogeographical group (NR: North relict; SR: South relict; NC, SC, and CC represent North, South, and Center cosmopolitan, respectively). North relicts showed higher directional selection to fecundity-seed mass trade-offs in comparison to the other groups. **C** Pairwise $F_{STQ}$ / $F_{ST}$ ratio comparisons between South relict and North relict and cosmopolitan groups using 1% top-SNPs related to fecundity, plant height, seed mass, fecundity response to water stress and intraspecific competition. The significance of the $F_{STQ}$ / $F_{ST}$ ratio was tested with a linear model through comparing $F_{ST}$ in non-coding regions versus $F_{ST}$ in top-SNPs of a trait. Student's *t* tests were applied to detect significant differences in $F_{STQ}$/$F_{ST}$ ratio for each trait between the South relict and the rest of the biogeographical groups. NS: not significant; *: *P* < 0.05; **: *P* < 0.01; ***: *P* < 0.001. The exact *P*-values can be seen in the 'Source Data file'.

reported[40,53], and was henceforth termed the relict group. Following the ADMIXTURE analysis, we classified the 71 genotypes into five biogeographical groups, considering the intersection of geographical origin and genetic clustering (Fig. 1A and Supplementary Table 1). Accordingly, there was a single group in the center: the 'Center cosmopolitan' group (*n* = 23); but two groups in both the south area ('South cosmopolitan', *n* = 17, and 'South relict', *n* = 7) and the north area ('North cosmopolitan', *n* = 13, and 'North relict', *n* = 11) (Fig. 1A).

Seed mass, fecundity, and plant height varied significantly between biogeographical groups (all *P* < 0.001; 0.17 < marginal-R² < 0.34; Fig. 1B–D; Supplementary Fig. 4; Supplementary Table 2). Specifically, we observed that Center cosmopolitans had 3-fold higher fecundities and produced significantly lighter seeds than South and North relicts (Fig. 1B, C). Moreover, Center cosmopolitans had an average height of 12 cm taller than the South relicts, but it did not display a significant difference from the North relicts (Fig. 1D). In between, South and North cosmopolitans exhibited a similar seed mass and plant height to Center cosmopolitans, except for their intermediate values of fecundity that spanned between the peripheral and central groups (Fig. 1B–D). In addition, fecundity measurements of plant response to competition and water stress', respectively, hereafter) revealed that the fecundity of Center cosmopolitans significantly decreased under competition compared to peripheral ones (up to 3-fold reduction), but not under water stress (Fig. 1E, F and Supplementary Table 2). Comparison between observed mean trait values and values obtained by resampling within biogeographical groups

confirmed that the phenotypic patterns were not the result of chance (Supplementary Fig. 5).

## Phenotypic variation is associated to the competition-colonization trade-off

Across the 71 genotypes under control conditions, fecundity was negatively correlated with seed mass (Pearson's *r* = −0.51, *P* < 0.001; Fig. 2A and Supplementary Fig. 6). Interestingly, this trend was also observed within some biogeographical groups, such as South relicts and North cosmopolitans (Supplementary Fig. 7). Across the 71 genotypes, fecundity under control conditions was also positively correlated with plant height (*r* = 0.51, *P* < 0.001; Supplementary Fig. 6) and negatively correlated with the fecundity response to competition (*i.e.* competition tolerance; *r* = −0.98, *P* < 0.001; Supplementary Fig. 6) and to water stress (*i.e.* water stress tolerance; *r* = −0.54, *P* < 0.001; Supplementary Fig. 6). Notice that stress response was calculated based on fecundity measurements under water limitation or competition, thus fecundity and stress response variables were non-independent.

Seed mass was positively correlated to the fecundity response to competition (*r* = 0.45, *P* < 0.001; Supplementary Fig. 6), but not to water stress (*r* = 0.12, *P* = 0.30; Supplementary Fig. 6). We also found that a large part of phenotypic variance was due to genetic variability, as measured by broad-sense heritability (0.40 < *H²* < 0.63; the lowest value corresponding to fecundity under competition and the highest value to seed mass; Supplementary Table 3). Together, these results suggest that phenotypic diversity in *A. thaliana* is influenced by a

trade-off between maximizing fecundity and dispersal (numerous light seeds and tall inflorescences) or maximizing competitive ability and stress tolerance.

To examine if the observed trait differences between groups were explained by natural selection, we estimated the quantitative genetic differentiation ($Q_{ST}$) of traits among biogeographical groups and compared it to the distribution of allelic differentiation along the genome among biogeographical groups ($F_{ST}$). The $Q_{ST}$-$F_{ST}$ approach aims to compare the degree of phenotypic differentiation (*i.e.* $Q_{ST}$) to the degree of genetic differentiation at neutral markers (*i.e.* $F_{ST}$) between populations[57,58]. The $Q_{ST}$ values of seed mass, fecundity, plant height, and the response of fecundity to competition were substantially extreme in the distribution of $F_{ST}$ across the genome (Supplementary Fig. 8), suggesting that these traits are under directional selection.

Using the method of Ovaskainen and colleagues[59,60], we further examined $Q_{ST}$-$F_{ST}$ in a multi-trait framework to test whether biogeographical groups have diverged more strongly in quantitative traits than expected solely due to genetic drift. This method estimates a statistic, denoted as $S$, which describes the phenotypic divergence of trait pairs among groups resulting from drift only ($S = 0.5$), directional selection ($S = 1$), or stabilizing selection ($S = 0$). The divergence among groups in the relationship between seed mass and fecundity was high and explained by directional selection ($S = 0.92$). Specifically, we found that the North relicts exhibited the highest divergence along the relationship between seed mass and fecundity (Fig. 2B). Likewise, we showed that divergence among groups in the relationship between height and seed mass, as well as between fecundity and height, was explained by the effects of directional selection ($S = 0.94$ and $S = 0.83$, respectively). Finally, we also observed high $S$ values when examining the divergence in covariations between fecundity under control vs fecundity under competition ($S = 0.82$), as well as fecundity under control vs fecundity under water stress ($S = 0.81$). This pattern extended to large $S$ values for the covariations between fecundity under control and the fecundity response (*i.e.* the difference of fecundity to water stress, $S = 0.80$, and to competition, $S = 0.81$, respectively). Together, these results suggest that trait differences between biogeographical groups cannot be only explained by neutral processes such as genetic drift.

## Genetic convergence between peripheral populations at genes involved in dispersal and competitive ability

The large sequencing effort of *A. thaliana* has provided an opportunity to identify the single nucleotide polymorphisms (SNPs) involved in trait variation through genome-wide association studies (GWAS)[61]. No significant SNP associations were found here, which could be attributed in part to the limited number of genotypes used for GWAS ($n = 71$), resulting in reduced statistical power, and part to the polygenic nature of the traits used here. Using a polygenic GWAS approach[61], we calculated the quantitative effect of 474,708 SNPs along the genome on each studied trait. At the whole-genome level, we observed negative correlations between SNP effects on fecundity in control conditions and SNPs related to seed mass ($r = -0.01$, $P < 0.001$; Supplementary Fig. 9A), as well as between SNP effects on the fecundity in control and SNPs related to fecundity responses to competition and water stress ($r = -0.28$, $P < 0.001$ and $r = -0.07$, $P < 0.001$, respectively; Supplementary Fig. 9A). In addition, SNPs related to plant fecundity in control also had a positive effect on plant height ($r = 0.08$, $P < 0.001$; Supplementary Fig. 9A). Overall, this suggests that the trade-offs observed between traits related to dispersal, stress tolerance, and competition tolerance are supported by pleiotropic effects of genes influencing trait values. We then extracted the top 1% (*i.e.* 4748) SNPs evenly distributed between the strongest positive and negative effects on each trait (*i.e.* 0.5 % of the SNPs having the most negative effects, and 0.5% of the SNPs having the most positive effects). Interestingly,

we observed similar directions in the correlations between SNP effects linked to traits using the 1% top-SNPs and the whole-genome SNPs (Supplementary Fig. 9A, B). The strongest negative correlations were found between 1% top-SNP effects on fecundity and plant height and SNP effects on fecundity response to competition and water stress ($-0.29 < r < -0.46$; $P < 0.001$; Supplementary Fig. 9B); and the strongest and significant positive correlations were found between SNP effects related to competition response and seed mass ($r = 0.33$; $P < 0.01$; Supplementary Fig. 9B) and competition response and water stress response ($r = 0.35$; $P < 0.001$ Supplementary Fig. 9B). The magnitude of the correlation coefficients was higher for the 1% top-SNPs than at whole-genome level (Supplementary Fig. 9A, B). Furthermore, many SNPs from the 1% top-SNPs were common to different traits, in particular for fecundity-related traits (Supplementary Fig. 10). However, some correlations were found significant at the whole-genome level but not among 1% top-SNPs (seed mass vs fecundity and plant height, and fecundity vs plant height).

To determine the genomic level $Q_{ST}$, we calculated the SNP-level $F_{ST}$ between geographical groups using different subsets of top SNPs (0.5%, 1%, 2%, and 5%) for each trait. This measurement allowed us to estimate trait $F_{STQ}$ (*sensu* ref. 58), contrasting it with $F_{ST}$ at neutral markers derived from non-coding SNPs. At all cutoff values, the $F_{STQ}$/$F_{ST}$ ratio, *i.e.* the divergence of quantitative traits from neutral molecular markers at the genomic level, for seed mass revealed that this trait diverged between all biogeographical groups, more strongly between South relicts and all cosmopolitan groups (Fig. 2C, Supplementary Figs. 11, 12 and Supplementary Table 4), which pointed again to a strong effect of directional selection on seed mass. Specifically, pairwise comparisons of $F_{STQ}$/$F_{ST}$ ratio between biogeographical groups showed that South relicts were comparatively (at genes related to seed mass) closer to North relicts, and farther from Center cosmopolitans (Fig. 2C). The $F_{STQ}$/$F_{ST}$ ratio for other traits, including fecundity, plant height and the fecundity response to competition and water stress, were relatively close to 1 (Fig. 2C). Interestingly, pairwise comparisons of the $F_{STQ}$/$F_{ST}$ ratio for plant height among geographical groups revealed a significant genetic distance between South and North relicts compared to Center cosmopolitans at the 1%, 2%, and 5% top-SNPs thresholds (Supplementary Figs. 11, 12 and Supplementary Table 4). Conversely, while no significant genetic disparities in fecundity were observed between South relicts and Center cosmopolitans, notable differences were apparent between North relicts and Center cosmopolitans (Supplementary Fig. 12 and Supplementary Table 4).

The genetic relatedness between some northern and southern genotypes has been explained by their higher abundance in common outlier haplotypes (measured through 10 kb windows along the genome[40]). Accordingly, the 71 genotypes used here displayed a similar pattern of genome-wide increase in outlier haplotypes at both north and south margins (Supplementary Fig. 13). We then examined whether the 1% top-SNPs involved in dispersal, competition, and stress tolerance traits were enriched or impoverished (compared to a random subset of non-coding SNPs representative of the genome-wide variation) in outlier haplotypes, particularly in populations located near the range margins. There was a strong enrichment of outlier haplotypes in SNPs having a positive effect on seed mass for South and North relicts. Conversely, these same SNPs were depleted in outlier haplotypes in cosmopolitan groups (Fig. 3A, B). On the contrary, the SNPs hurting seed mass were significantly depleted for outlier haplotypes in North relicts, while they were enriched in outlier haplotypes in South, Center, and North cosmopolitans (Fig. 3C, D). Surprisingly, we observed that relicts had enrichment of outlier haplotypes with a negative effect on seed mass, but it represented a percentage substantially lower than the percentage of enrichment of haplotypes with a positive effect (0.6% vs. 8.5%, Fig. 3A, C), perhaps reflecting the high diversity within the relict group. The percentage of outlier haplotypes

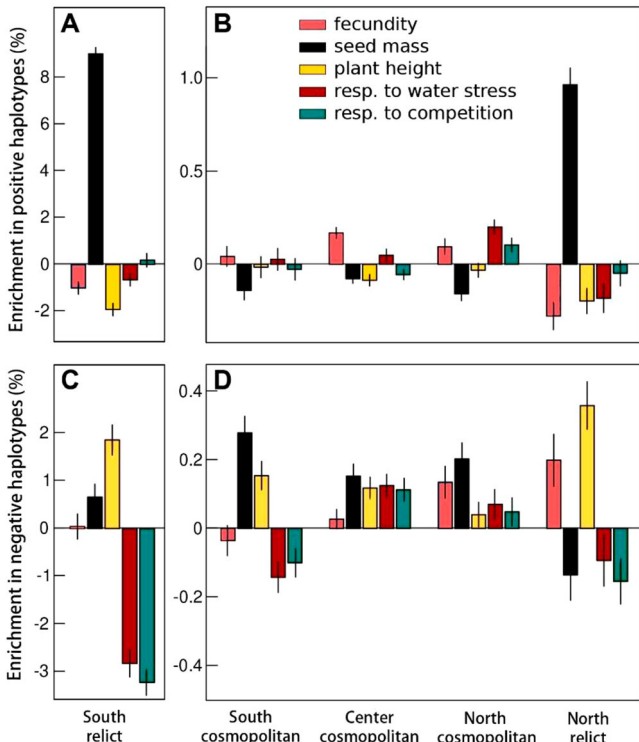

**Fig. 3 | Enrichment in relict haplotypes (%) among biogeographical groups.**
Enrichment (%) in outlier (*i.e.* relict) haplotypes with a positive (**A** and **B**) and negative (**C** and **D**) effect on each trait studied for each biogeographical group. A bootstrap approach was applied with 50 permutations, each taking 1000 SNPs among the 0.5% positive effect and others 1000 SNPs among the 0.5% negative effect on a trait, and 1000 SNPs randomly chosen among non-coding SNPs (to represent the neutral genomic background). The enrichment was calculated as the ratio between the proportion of outlier haplotypes in top-SNPs and those in non-coding regions. Data are presented as mean values +/− SEM.

with a positive effect on fecundity and plant height decreased at both opposite margins, together with the enrichment of outlier haplotypes hurting fecundity in North relicts. Similarly, the percentage of outlier haplotypes with a negative effect on stress and competition responses decreased at both the opposite margins (Fig. 3C, D). Together, these results suggest a strong genetic convergence between South and North relicts in genes related to dispersal traits such as seed mass and fecundity, as well as in genes associated with competition and stress responses.

## Discussion

In evolutionary ecology, a significant focus is directed towards understanding the factors influencing plant species distribution and the dynamics of their populations[62]. Given that every species faces the challenge of coexistence with others[63], many studies have focused on interspecific interactions to explain species distribution[64,65]. Specifically, the trade-off between competitive and colonization abilities is considered as a fundamental mechanism for species coexistence[66,67]. However, for species that live and disperse their seeds in patches, neighbors are more likely to be conspecifics than representatives of other species[20,22]. Consequently, intraspecific competition and colonization mechanisms are thought to significantly influence population dynamics. Despite this, only a limited number of studies have investigated competition/colonization relationships at the intraspecific scale[36,37], and even fewer have considered them from a biogeographical perspective. In this study, we investigated the intraspecific variability of competition- and colonization-related traits in European

populations of *Arabidopsis thaliana*, shedding light on the species' demographic and evolutionary history. We observed a clear differentiation in traits such as seed mass, fecundity, plant height, and competition response between peripheral and central populations. These variation patterns directly align with recent genetic findings about the history of *A. thaliana*[39,40]. Cosmopolitan genotypes represent 95% of the species' European diversity, resulting from the demographic success of a former cosmopolitan group originally from the Balkans or Caucasus[40,41]. This group outperformed all others, occupying central Europe until that point. One of the main hypotheses explaining this demographic success suggested that genotypes from this central group were more competitive[40]. This assumption is likely derived from the theoretical peripheral-core hypothesis, which proposes that dispersal ability would be more advantageous in peripheral populations for favoring colonization of novel environments and range expansion. Conversely, competitive ability might be expected to prevail in central populations[68]. Instead, our results suggest that central cosmopolitan genotypes exhibit a highly ruderal strategy[3] characterized by a high fecundity, light seeds, and tall inflorescence. This trait combination might have enabled genotypes from the Balkan group (i) to avoid intraspecific competition with neighboring seedlings, and thus have a greater chance of survival[69] and/or (ii) to disperse further and thus colonize new patches of vegetation[36,37]. This latter aligns with François and colleagues' hypothesis[56] about the parallelism between the entrance of the cosmopolitan eastern European source of *A. thaliana* and the creation of a dynamic and perturbed landscape with the diffusion of agriculture starting from the east of Europe. Supporting this argumentation, Williams and colleagues[36] demonstrated a more than threefold increase in dispersal distance for evolved *A. thaliana* populations in highly-fragmented landscapes after six generations. Moreover, they showed that the coefficient of variation for spread was four times greater in the patchiest landscapes (12 times the mean dispersal distance) than in the continuous landscapes.

Many studies have revealed consistent patterns in the traits of *Arabidopsis thaliana* populations along the European latitudinal gradient, such as flowering time, plant height, and rosette size[70,71]. These variations are likely influenced by environmental factors associated with latitude, including temperature and day length. For example, populations at higher latitudes may exhibit delayed flowering to avoid late spring frost, while those at lower latitudes may have adaptations for coping with higher temperatures[44]. Here, we observed a center-to-margin trait differentiation rather than latitudinal clines, which was reflected in U-shape, or inverse U-shape, relationships between traits and latitude (Supplementary Fig. 2). Colonization-related traits decreased from central toward both northern and southern peripheral groups, illustrated by the reduction in fecundity values and, to a lesser extent, plant height. This differentiation between the north and south was explained by the presence of relict genotypes at the opposite margins. For instance, peripheral groups (South and North relicts) were more competitive, *i.e.* higher seed mass values, than their central counterparts. Consistent with these findings, Estarague and colleagues[9] recently showed phenotypic differentiation in leaf traits associated with competitive ability (leaf area, C-scores in CSR strategy) between central and north/south *A. thaliana* populations across Europe. Moreover, Clauw and collaborators[72] showed that northern peripheral genotypes took advantage of their big seeds to deal with cold temperatures and survive winter stress. Their research revealed that big seeds in northern populations were associated with lower growth rates, potentially aiding survival in harsh environments. This is consistent with other studies indicating a prevalence of slow-growing and long-lived plants at the geographical margins, contrasted by the prevalence of fast-growing and short-lived species in central habitats[46,73]. While studying traits offers essential explanatory hypotheses, the next challenge lies in experimentally testing the greater dispersal ability of

cosmopolitan populations and assessing the subsequent performance gain.

Beyond uncovering the evolutionary and ecological history of *A. thaliana*, our study shows evidence of the competition-colonization trade-off as a mechanism structuring phenotypic divergence at the intraspecific level. As a plant model species, *A. thaliana* offers a wealth of molecular background information to delve into evolutionary mechanisms that shape phenotypic relationships. Here, polygenic GWAS analysis evidenced that SNPs positively linked to colonization function had a pleiotropic and negative effect on fecundity response to competition and water stress (see SNPs correlations on the involved traits; Supplementary Fig. 9)[74]. In other words, our study suggests that both colonization and competition functions could not be selected for the same genetic background. Other plant studies have also evidenced that the competition-colonization trade-off is under selection. For example, Fakheran and collaborators[37] found *A. thaliana* populations grown at high-density and in disturbed environments diverged in both competitive and dispersal abilities in only five generations.

Our study highlighted that seed mass is a key adaptive component to survive and persist in peripheral areas[26,72,75,76]. Indeed, we detected a high difference in $F_{STQ}$-$F_{ST}$ ratio between groups for seed mass, as well as an enrichment of outlier haplotypes with a positive effect on seed mass in both South and North relicts. Furthermore, we observed an increased occurrence of outlier haplotypes negatively influencing stress tolerance traits (competition and water stress) as we moved from peripheral to central groups. In other words, less adaptability to stressful conditions for Center cosmopolitans is also demonstrated under intraspecific competition through phenotypic evidence. In contrast, Center cosmopolitans showed the highest value in enrichment of outlier haplotypes with a positive effect on fecundity; but not on plant height as we expected. Plant height is a major trait related to dispersal distance[77,78]; however, it can be constrained by many other unmeasured aspects of species life-history strategies like species longevity[79]. Interestingly, South and North cosmopolitans showed an intermediate pattern between central and peripheral, both at the phenotypic (mainly fecundity) and genetic levels. These results suggest a potential introgression of outlier haplotypes from groups with a high amount of outlier haplotypes (south and north relict groups) to cosmopolitan groups in areas where both coexist (*e.g.*, the Iberian Peninsula and Sweden). Admixture and genetic introgressions between groups have already been documented in *A. thaliana*. This may have allowed more recent colonizers to obtain locally adaptive alleles, aiding their survival and persistence in regions where the species faces environmental constraints[46,54,56]. Consistently, evidence of genetic admixture between relict and cosmopolitan groups has been found in genes linked to flowering traits and resource-use strategies[55]. However, the extent to which such hybridization might contribute to local adaptation in contrasting environmental conditions warrants further in-depth investigation.

## Methods
### Plant material and genotype classification
We used a total of 71 natural genotypes of *A. thaliana* from three geographical areas representative of center, south, and north Europe, which are separated by natural barriers that partially isolate plant populations. We used the latitude threshold of 45°, which corresponds to the Pyrenees and Alps mountains, to delineate the south area, and the latitude threshold of 55° that separates the Scandinavian Peninsula (Fig. 1A). We excluded genotypes originating from sites with an altitudinal distribution above 1000 m.a.s.l. to avoid confounding factors associated to elevation (Supplementary Table 1). All genotypes were included in the initial germplasm of the 1001 Genomes project (http://1001genomes.org/[39]), and seeds were supplied by the Nottingham *Arabidopsis* Stock Center (NASC) and the *Arabidopsis* Biological Resource Center (ABRC).

We downloaded full genomic sequences from the 1001 Genomes Project available and filtered SNPs with minor allele frequencies (MAF) superior to 5% among the 71 genotypes. Genetic clustering was performed with ADMIXTURE[80,81] after linkage disequilibrium pruning ($r^2 < 0.1$ in a 50 kb window with a step size of 50 SNPs) with PLINK[82], resulting in 47,213 independent SNPs used for subsequent analyses. A cross-validation (CV) for different numbers of clusters ($k = 1$ to $k = 9$) showed that the set of studied genotypes was best separated into two groups ($k = 2$ presented the lowest CV error = 1.00, Supplementary Fig. 3). Following the same approach as the 1001 genomes project[39], we assigned each genotype to a group if more than 50% of its genome derived from the corresponding cluster. Thus, we classified the 71 initially selected genotypes into five biogeographical groups, considering the interplay between geographical origin and genetic clustering (Fig. 1A and Supplementary Table 1). There was a single group in the center: the 'Center cosmopolitan' group ($n = 23$); and two groups in the south area ('South cosmopolitan', $n = 17$, and 'South relict', $n = 7$) and in the north area ('North cosmopolitan', $n = 13$, and 'North relict', $n = 11$).

### Greenhouse experiment and treatments
The experiment was carried out in two adjacent compartments of a greenhouse, where plants were sown and grown in individual pots ($7 \times 7 \times 6.5$ cm) filled with a 1:1 mixture of commercial peat moss (Neuhaus N2) and vermiculite (medium grain 3–6 mm). In the control treatment (non-stressful conditions), three to eight seeds per pot were sown, and the first germinated plant was kept and grown at the pot center under well-watered conditions (regular irrigation every 4-5 days during the experiment). In the intraspecific competition treatment, we sowed between 3-4 seeds at the pot center and at the four-pot corners to assure a germinated plant in each position. If several seeds germinated per position, we kept the only first germinated one and the others were thinned. Therefore, a focal *A. thaliana* plant was grown at the pot center surrounded by four individuals of the same genotype, under similar well-watered conditions as the control treatment. In the water stress treatment, the first germinated plant was selected at the pot center, *i.e.* without competing neighbors, and grown in water stress conditions – regular irrigation every 10-11 days along the experiment since plants reached three - four true leaves (Supplementary Fig. 1). For the first 40 days from sowing (from end of January to mid-March 2020), we kept all plants in cold temperature (~10 °C) under well-watered conditions to ensure establishment and vernalization of all genotypes (*i.e.* breaking the genetic suppression of flowering). After the vernalization period, we raised the room temperature (20 °C day / 15 °C night) and applied different watering treatments. Plants were consistently exposed to the natural light-dark cycle from sowing to the end of the experiment.

We used eight pot replicates of each genotype and treatment, resulting in a total of 71 genotypes x 8 replicates x 3 environments = 1704 pots (Supplementary Fig. 1). Plants were equally distributed in the two adjacent compartments, with three large tables each containing one treatment per table and compartment. Each table was divided into four similar blocks, that included one replicate per genotype randomly placed. The tables were rotated within the compartments and turned around themselves every two days to minimize putative microclimate heterogeneity in each compartment.

### Trait measurements
We harvested plants as each one reached maturity, *i.e.* at plant senescence when fruits started to dry (from mid-April to mid-August 2020), to avoid bias in resource allocation in vegetative or reproductive organs between late and early flowering plants, and then performed a set of phenotypic measurements. For each focal individual, we measured the maximum reproductive height (cm) from the rosette base to the apex of the longest flowering stem[77]. We counted the total

number of fruits produced for each focal individual and measured the average fruit length from four fruits chosen randomly along the main inflorescence. We then calculated plant fecundity by multiplying the total number of fruits at the whole-plant level by the average fruit length to generate an estimate of the total number of seeds per plant[83]. On the other hand, we measured the seed mass (BALCO MC5, mg) of focal individuals ($n = 4$-5 per genotype) by weighing air-dried seeds ($n = 10$-30 per individual) and then dividing the total air-dried weight by the number of seeds in the sample. Finally, we measured the fecundity response to competition or water stress as the absolute difference in the mean fecundity of genotype $i$ in stress treatment (intraspecific competition or water stress) and mean fecundity of genotype $i$ in control conditions. Although there is an active debate in the literature about the use of relative versus absolute fitness, with no clear consensus[84–88], we consider that this is the absolute "amount of progeny lost" that is relevant for population demography in an evolutionary perspective since it might impact the demographic success of a given genotype. Notice that there is a lack of independence between fecundity and stress response traits, which is reflected with a high correlation between both (Supplementary Fig. 6).

### Statistical analyses

**Phenotypic analyses.** From the initial 1,704 individuals, we finally conducted analyses over a total of 1595 individuals, after discarding plants that did not complete their life cycle at the end of the experiment or died during the experiment. In total, 544 plants grew under control, 507 grew under intraspecific competition, and 544 were under water stress conditions. In the competition treatment, we did not consider in our analyses the focal plants with the absence of one or more of their four neighbors to avoid potential bias linked to neighboring plant density. Further, we removed 32 individuals due to human errors in plant identification and 23 individuals, which were truly measured but showed extreme values within genotype and treatment after applying the Hampel filter, *i.e.* the median, plus or minus 3 median absolute deviations[89,90].

We first ran genotypic mean correlations among measured phenotypic traits, without the assumption of causal effect. To assess differences in plant dispersal, colonization, and competition abilities among *A. thaliana* geographical groups, we ran linear mixed-effects models for plant height, fecundity, and seed mass on individuals in control conditions. We used the geographical group as a fixed factor and genotype identity (nested within the geographical group) and blocks (nested in a table and these, in turn, nested in the compartment) as random factors. With the same fixed structure, we ran linear models to quantify differences in genotypic mean responses in fecundity to intraspecific competition and water stress among geographical groups. When the variable geographical group was significant, we contrasted mean phenotypic differences among geographical groups through *post hoc* Tukey's tests. We also plotted each genotypic mean trait against latitude using loess regressions. Moreover, we ran null models on genotypic mean trait by randomly shuffling labels (100 replicates) on biogeographical groups for each genotype, while keeping sample sizes fixed for each geographical group. We then plotted the observed vs null mean trait values for each genotype to test whether a phenotypic pattern can be considered by chance or not. All analyses were performed in R v. 3.5.1[91], using lme4[92] and emmeans[93] R packages.

**Genetic analyses.** For all traits, we ran Bayesian sparse linear mixed models (BSLMM) implemented in the software package GEMMA[54]. BSLMM is a polygenic model that assesses the contribution of multiple SNPs to phenotypic variation, accounting for relatedness via the inclusion of a kinship matrix as a covariate. From the BSLMMs, we obtained a dataset with 471,453 SNPs, which was subsequently used to estimate the join effect of these SNPs on each studied phenotypic trait. The total effect size $E_i$ of each SNP$i$ on every single trait was determined as $E_i = \alpha_i + \beta_i * \delta_i$, where $\alpha_i$ corresponded to the estimate of small SNP effects, $\beta_i$ corresponded to the estimate of large SNP effects, and $\delta_i$ corresponded to the probability of non-zero effect of a locus when accounting for the effects of all other loci[54,94]. We computed a linear correlation matrix between all SNP effects $E_i$ associated with each phenotypic trait to assess whether ecological trade-offs were embedded at the genetic level.

To test whether phenotypic differentiation among geographical groups departed from neutral processes, we compared the structure of genetic variance of traits among populations ($Q_{ST}$) to their neutral genetic differentiation. To proceed, we first estimated Weir and Cockerham $F_{ST}$[57] between each pair of geographical groups for all SNPs across the genome using PLINK software[82]. $Q_{ST}$ of each trait was then calculated as the ratio of genetic variance of each trait within and among geographical groups from genotypic means. We tested if $Q_{ST}$ values were extreme in the distribution of $F_{ST}$ in a bootstrapped and non-parametric approach. Second, we tested in a multi-trait framework if the divergence of trait values across populations was merely explained by drift or natural selection by using the method of Ovaskainen and colleagues[59]. This method estimates a statistic $S$, which describes the phenotypic divergence of trait pairs among groups by drift only ($S = 0.5$), by directional selection ($S = 1$), or by stabilizing selection ($S = 0$)[95]. To do so, we first estimated a neutral co-ancestry matrix with RAFM[96] package, by using 22,307 non-coding SNPs (*i.e.* outside genes), for which at least 95% of genotypes were genotyped. We then used DRIFTSEL package[60] to compare the phenotypic divergence of pairs of traits to genetic neutral divergence. We fitted both the RAFM and DRIFTSEL models with 5000 MCMC iterations, discarded the first 1000 iterations as transient, and thinned the remaining by 4 to provide 1000 samples from the posterior distribution. Third, we extracted from BSLMMs the 0.5% SNPs with the strongest positive effect and the 0.5% SNPs with the strongest negative effect on every trait, making a total of the 1% top-SNPs (i.e. 4,748 SNPs per trait). The $F_{STQ}$ parameter was estimated for each trait as the mean $F_{ST}$ of the 1% top-SNPs for the given trait ($F_{STQ} / F_{ST}$)[97]. We estimated the significance of $F_{STQ} / F_{ST}$ ratio with a *Student's t* test ($F_{ST}$ in non-coding regions versus $F_{ST}$ in top-effect-SNPs of a trait). Finally, we quantified the $F_{STQ} / F_{ST}$ ratio differences between all cosmopolitan groups and the 'South relict' group for every trait. In addition, we tested the sensitivity to changes in our $F_{STQ} / F_{ST}$ ratio depending on top-SNPs cutoff value. To do so, we re-calculated the $F_{STQ} / F_{ST}$ ratio using three different top-SNPs cutoffs values, 0.5%, 2%, and 5% top-SNPs, for every trait. Finally, we calculated the broad-sense heritability ($H^2$) for each study trait with a linear mixed-effects model, considering the genotype identity as a random factor. $H^2$ was calculated at the proportion of genotypic variance ($\sigma^2_G$) over the total variance ($\sigma^2_G + \sigma^2_E$).

**Enrichment of outlier haplotypes.** Using the calculation of the genomic proportion of relict ancestry for a 10 kb window for each genotype previously quantified by Lee and collaborators[40], we first plotted the number of outlier haplotypes for each one of our geographical groups and then estimated the proportion of outlier haplotypes among our selection of 1% top-SNPs with the strongest positive and negative effect on each studied trait, and among SNPs randomly sampled in non-coding regions. We applied a bootstrap approach with 50 permutations, each taking 1000 SNPs among the 0.5% positive effect and others 1000 SNPs among the 0.5% negative effect on a trait, and 1000 SNPs randomly chosen among non-coding SNPs (to represent the neutral genomic background). Then, we estimated the average proportion of outlier haplotypes for these three categories (top-positive, top-negative, and non-coding) among all genotypes within each geographical group. The 'enrichment in outlier haplotypes (%)' for every trait and geographical group was calculated as the difference between the proportion in outlier haplotypes in top-SNPs and those in non-coding regions.

## Reporting summary

Further information on research design is available in the Nature Portfolio Reporting Summary linked to this article.

## Data availability

The mean functional trait and genetic data generated in this study have been deposited and are freely available in the Figshare Digital Repository database under accession code https://doi.org/10.6084/m9.figshare.23807346.v2[98]. Accessions used in this study had been previously described by the 1001 Genomes project (http://1001genomes.org/) and their codes can be seen in the Supplementary material file (Supplementary Table 1). Source data are provided as a Source Data file. Source data are provided with this paper.

## Code availability

The codes supporting the plots within this paper and other study findings are available from the corresponding author upon reasonable request.

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

## Acknowledgements

We are very grateful to Estelle Leca, Estela Illa Bachs, and Stefania Przybylska for their help in the setting of the experiment. We also thank Maylis Combes, Anais Hany, and Thibault Martino for their valuable help in measuring phenotypic traits. We thank the technical platform "Plateforme des Terrains d'Expérience du LabEx CeMEB" in Montpellier for their support and advice during the experiment. This work was supported by CNRS, INRAE, the French Agency for Research (ANR grant ANR-17-CE02-0018-01; "AraBreed" to F.V., D.V., and C.V.), and the European Research Council (ERC;"CONSTRAINTS": grant ERC-StG-2014-639706-CONSTRAINTS to C.V.). This work was also supported by the Ramón Areces Foundation and, the Junta de Andalucia (Spain) and the European Social Fun 2014-2020 Program (DOC_01035) to CCB. Also, the Office of the Director of the National Institutes of Health's Early Investigator Award with award number: 1DP5OD029506-01; by the U.S. Department of Energy, Office of Biological and Environmental Research, grant number: DE-SC0021286; and by the Carnegie Institution for Science, supported to M.A.-E. This work was supported by the European Research Council (ERC) ('PHENOVIGOUR': grant ERC-StG-2020-949843 to FV).

## Author contributions

A.E., C.C.B., D.V., F.V., and C.V. conceived and designed the experiments. C.C.B. and A.E. performed the common garden experiment. C.C.B., A.E., and D.V. carried out the lab measurements. C.C.B., F.V., and A.E. analyzed the data. C.-R.L. provided genomic data about the proportion of relict ancestry of each genotype and together with M.E.-A. gave support with statistical analyses. C.C.B., A.E., E.G., and F.V. wrote the paper with comments from all authors.

## Competing interests

The authors declare no competing interests.
