## [Peer Review File · Nature Communications]

Ecological trade-offs drive phenotypic and genetic differentiation of *Arabidopsis thaliana* in EuropeREVIEWER COMMENTS

Reviewer #1 (Remarks to the Author):

This paper seeks to examine the evolutionary ecology of plant dispersal using the model plant *A. thaliana*. The use of *A. thaliana* in part can provide genetic insights into the process. The authors focus their study on geographic population of *Arabidopsis*, particularly those that are considered relict or nonrelict populations associated with the last glaciation. They examine several phenotypes in the framework of the colonization-competition trade-off (or colonization-stress tolerance trade-off) model, and use SNP data to look at the polygenic basis for these traits.

In general, the paper looks at an important issue in plant evolutionary ecology, and the hypotheses and framework of the authors are good. The phenotype data is convincing with regards to the trade-offs observed (although this is not that surprising given results in other plant species). The genetic data suggesting that there is selection for SNPs associated with competition/colonization traits is also very interesting.

The issue I see is that the study does not go far enough to merit publication in this journal. The results (particularly the genetic data) is tantalizing, but to merit publication in a general science journal I would have hoped to see more genetic dissection of the key ecological traits. Granted their framework is that these traits has a polygenic basis, but stronger functional genetic information coupled with stronger evidence of genetic selection will be necessary to warrant publication in a general science journal rather than a more specialist journal.

Reviewer #2 (Remarks to the Author):

This paper addresses an interesting and important question: Can tradeoffs between dispersal, competitive ability, and stress tolerance explain the distribution of *Arabidopsis thaliana* life history phenotypes across Europe? Specifically, can the recent spread of “non-relict” genotypes as human commensals across central Europe be explained by their dispersal ability, at the cost of stress tolerance and competitive ability which were favored at the northern and southern range edges? The study’s combination of experimental common garden data with genomic analyses is commendable. The common garden was well designed and executed, although I wish it had been possible to include more accessions. Unfortunately, however, I had a lot of difficulty understanding the methods and have concerns about some of the analyses, especially of reaction norms to stress. Overall, I’m not entirely convinced that the results as presented justify some of the conclusions. Nevertheless, if the authors can address the issues below, this study could be a very useful and interesting contribution.

Specific comments:

Introduction:

Lines 74 – 78 I was surprised that the authors don’t cite relevant experimental evolution papers by Fekheran et al (2010, PNAS) and Williams et al. (2016, Science), which both use the Ler x Cvi *A. thaliana* RILs (a non-relict x relict cross) to examine evolution of seed mass, competitive ability, and/or dispersal traits. The Fekheran paper in particular addresses competition/colonization tradeoffs and the implications for the current results should be mentioned in the discussion.

Line 96. Is the 10,000 year time frame generally accepted ? See Fulgione et al. 2018.

Fig. 1. The dark green region representing the origin of the second expansion is not the same as the origin shown in Figure 5 of Lee et al. 2017.

Line 127. There were only 71 accessions in the experiment. Say so here and throughout.

Lines 131-132: objective (iii) is very general and vague and could be sharpened.

Methods:

Lines 335-339: How many accessions were in each of the groups?

Line 371: When was maximum reproductive height measured? Were plants harvested and measured at senescence?

Lines 378- 381., Line 401. I have major concerns about how these "individual" reaction norms were calculated. Reaction norms to course-grained environments like stress and competition are properties of genotypes, not individuals. Consequently, it is inappropriate to run linear models on individual "reaction norms" for these genotypic traits. A simple model run on genotype plasticities calculated from genotypic means in the relevant treatments seems more suitable for comparing tolerances across geographic groups, but power is low. A better option might be to treat fecundity in control, competition, and water stress environments as three separate traits for Q_{st}/F_{st} and examine their multivariate divergence (e.g. Ovaskainen et al. 2011, Genetics). Bivariate plots of control vs treatment fecundities might be quite informative about reaction norm evolution. In any case I would like to see fecundities in competition and water stress presented as traits in figures 3, 4, S2, S3, and S4. I suspect that may turn out to be interesting tradeoffs in size and fecundity between control and stress treatments that would be worth exploring

Line 391. It's fine to eliminate outliers, but it would be nice to know whether these were biologically real values, or probable measurement errors.

Genetic analyses: 71 genomes are quite a small sample size for GWA, especially spanning a large geographic area with population structure. Were any significant peaks detected?

Enrichment of outlier haplotypes:

Line 440-441. "genomic proportion of relict ancestry" might be easier to understand. I had to go back to the Lee paper to be sure I understood what was done.

Lines 442-443. Why the top 1%? How sensitive were the results to changes in this cutoff value?

Results:

Lines 142- 145. This sentence is very confusing – it seems to be the result of a cut and paste error during editing. Also, outlier haplotypes need to be defined and explained briefly here, since most readers of a short format paper like this will read the methods last. The definition of outlier haplotypes in the caption of Fig.1 b is similarly confusing.

Figures 2 and S3. These are correlations, with no assumptions about causality, so regression lines should not be shown. In Fig S3, the figure would be easier to read if r values could be given within the same panels as the scatterplots rather than above the diagonal. Again, the correlations involving water and competition response should be calculated from reaction norms between genotype means in control and treatment, not from individual plants. And fecundities in competition and water stress should also be included as traits in Fig S3. Since fecundity was used to calculate responses to water stress and competition, these are not independent variables, and the observed correlations between them may in part be artifacts of the calculation.

Lines 188-191. Yes, these correlations are significant, but that's because of the huge sample size. Are they really biologically meaningful evidence of a genetic tradeoff? What would happen if you calculated correlations just with the top 1% of associated SNPs for each trait?

Line 196-197. Again, I don't think it is appropriate to calculate Q_{st} for stress response traits based on calculations of "individual" reaction norms. Q_{st} should be reported for fecundities in the three experimental treatments, and the Ovaskainen (2011) method for examining multivariate

divergence might be helpful.

Line 198 "structuration" should be "structure"

Lines 211-213. Given my concerns about the calculation of these variables, I'm not sure how to interpret these results. I would trust them more if the GWAS to identify top SNPs had been done on reaction norms calculated entirely from genotypic means, as well as separately for fecundity values in all the treatments. Also: why not also examine divergence among non-relict groups?

Discussion:

Lines 239- 240. Although I largely agree with the idea that populations may have differentiated along colonization/competition and colonization/tolerance axes, I think that this sentence overstates what can confidently be concluded from the results presented here.

One important question that I would like to see a little speculation about is how selection acted on colonization ability. Is this a case of selection for long-distance dispersal ability during the colonization process? Or is it due to selection for dispersal in metapopulations in ruderal disturbed environments? Or was selection on seed size relaxed in ruderal environments, removing constraints on selection response for higher seed number?

Line 242. Fulgione et al. 2018 could also be cited here.

Lines 249- 251. Is there evidence of this tradeoff within regions? Also, how much of the variation among populations can be explained by continuous variation in the environment of origin (e.g. land use type, climate). I really like Figure S5 and it might be nice to put it in place of Fig. 2 (with r and p values shown as in the original and regression line removed).

Lines 268- 269. I think this statement is an overreach given the tiny genomic correlation coefficients observed. Although significant due to the large sample size, they don't provide strong evidence of pleiotropy or genetic constraint. The authors might also consider the possibility that the observed phenotypic syndromes emerged through correlated multigenic selection on entirely independent loci underlying the different traits. How much overlap was there in the top SNPs for the different traits?

Lines 272- 273. Fakheran et al. (2010) and probably Williams et al (2016) could also be referenced here.

Figure S2. This figure would be more useful if it could be arranged according to geographic groups, or at least color-coding by group. Also please include fecundity in competition and stress treatments as traits.

Reviewer #3 (Remarks to the Author):

- What are the noteworthy results?

- Will the work be of significance to the field and related fields? How does it compare to the established literature? If the work is not original, please provide relevant references.

- Does the work support the conclusions and claims, or is additional evidence needed?

- Are there any flaws in the data analysis, interpretation and conclusions? - Do these prohibit publication or require revision?

- Is the methodology sound? Does the work meet the expected standards in your field?

- Is there enough detail provided in the methods for the work to be reproduced?

In this study the authors find evidence for SNPs associated with phenotypic trade-offs between colonization and competitiveness traits. These SNPs vary in frequency across *A. thaliana* range in a manner that might support the expectations for trade-off roles in species expansion, which has been rarely demonstrated empirically. The approach taken is novel and should be of interest to the evolution and ecology readership of the journal. However, I found very confusing to follow the narrative of the due to crossover of terms that do not necessarily means the same thing. Thus, I am not sure the conclusion are fair given the actual results, but possibly they are. There are two main issues that need clarification:

- 1) The geographical groups used and their meaning. "Central" versus "marginal" populations are the concepts used in the introduction, which implies that the relevance of trade off in limiting population expansion is going to be considered. However, the accessions sampled are from north, south and central Europe, and accessions are divided into "relict" and "non-relict". Is not clear how significant difference between the southern relict group from other groups actually translate to central or marginal populations and relate to theoretical expectations. Chuang & Peterson (2016) review indicate that theoretical expectations are of shifts toward greater reproductive output at some edge demes, which may aid in the establishment of a species in a novel environment. I don't think that is what was found here?

In the abstract it is said "We showed that a trade-off between plant fecundity and seed mass constrains the diversification of *A. thaliana* in Europe". But did the experiment tested directly whether relict genotypes can outcompete the cosmopolitan genotype? And is this equivalent to saying that there are differences between peripheral and central genotypes as stated in the abstract?

There was clear demonstration of trade-offs across accessions, and that it is likely this trade off has a genetic basis (measurements of heritability and genetic covariance would be helpful here), but how does that show that the trade-off constrained diversification? It would be helpful if the connections between theoretical expectations and the geographical groups connected are more clearly shown.

- 2) Population genetic models predictions and estimates are described as *fait accompli*. Much of the experimental design is relying on assumptions about the relationship between the geographical groups and how this relates to models of potential expansion of Arabidopsis; however these assumption need to be made more clearly. For example, I assume that the south to north choice of populations is based on the assumption that this is how Arabidopsis recolonized Europe after surviving in refuges in the Iberic peninsula. This would suggest though that the northern group is the “peripheral” or “marginal” group; but then the relicts and non-relicts comes on top of that, and it becomes quite difficult to match expectations and model estimates. This is further confused as the definition of relicts and their relevance is not entirely clear. But perhaps more importantly, is good to remember George Box’s famous quote: “All models are wrong, but some are useful”. There is indeed a good probability that recolonizations did happened as models predict... however, they are not “known” or “demonstrated”. The colonizations happened millions of years ago and we can only speculate that this is what happens. Thus, I suggest the language around much of the introduction and the discussion is toned down to reflect that all the population models for Arabidopsis colonization scenarios are not facts. For example is plausible that new lineages had to compete with older lineages, but it is also possible that there were new environments that new lineages moved in without no need to compete with any lineage. None of this invalidate the findings of the manuscript, which are interesting. It just should be presented in a more open minded way, i.e. if this is about range expansion, limitations of adaptations at the margins, there should be more objective ways to define “margins” than grouping into relicts and cosmopolitan genotypes.

There is overall an issue with clarity of approach and methods, and here is a non-exhaustive list of issues that could be clarified for the methodology and results to be better understood:

- How was the competition treatment established? Were seedlings transplanted into a pot? If so at what age? As written it seems like they were transplanted after coming out of the cold? That would be quite late in their life-history.
- Fecundity is calculated in centimeters? It is described as the multiplication of fruit number (including all branches, or just in the main branch? The latter could introduce many biases) by fruit length, but there is no unit associated with the variable in graphs and tables
- What is meant by the “reaction norm”(line 379)? Do you mean the slope of a regression is calculated? Is confusing, as it also says “difference” in fecundity of a particular individual? If is just the difference as in the subtraction of fecundity from the average fecundity in the control treatment, the term “norm of reaction” is not necessary or accurate. Furthermore, absolute differences will bias the estimate of competition towards accessions that produces more seeds, as the maximum competition would be constrained by the maximum fecundity. Perhaps using slopes would be better to remove such biases

- There is no definition of “outlier haplotypes” or how they were determined on the methods section before line 439; and no clarification on that section either. On the results section, there is a definition (line 143): “alleles less frequent across whole species genome”, but without knowing what is an allele (a single SNP, or a haplotype of a certain window size?) this terms remains unclear.
- Line 407 – I assume SNPs with frequencies higher than 5% were kept? Your sentence suggests the opposite
- Line 419 – I am unclear whether I understood this sentence. It is unclear what the “pair” is in “all pairwise combinations of SNP effects on the phenotypic traits ”?
- Line 422, results and discussion – The comparison of Q_{st} to F_{st} provides insight on whether there is departure from a neutral model. This may be caused by adaptation, but it can also be due to other causes, thus the use of the term “adaptive should be used with more caution.
- NASC stands for Nottingham Arabidopsis stock centre (line 341)

RESPONSE TO REVIEWERS' COMMENTS

Reviewer #1 (Remarks to the Author):

*This paper seeks to examine the evolutionary ecology of plant dispersal using the model plant *A. thaliana*. The use of *A. thaliana* in part can provide genetic insights into the process. The authors focus their study on geographic population of *Arabidopsis*, particularly those that are considered relict or nonrelict populations associated with the last glaciation. They examine several phenotypes in the framework of the colonization-competition trade-off (or colonization-stress tolerance trade-off) model, and use SNP data to look at the polygenic basis for these traits.*

In general, the paper looks at an important issue in plant evolutionary ecology, and the hypotheses and framework of the authors are good. The phenotype data is convincing with regards to the trade-offs observed (although this is not that surprising given results in other plant species). The genetic data suggesting that there is selection for SNPs associated with competition/colonization traits is also very interesting.

The issue I see is that the study does not go far enough to merit publication in this journal. The results (particularly the genetic data) is tantalizing, but to merit publication in a general science journal I would have hoped to see more genetic dissection of the key ecological traits. Granted their framework is that these traits have a polygenic basis, but stronger functional genetic information coupled with stronger evidence of genetic selection will be necessary to warrant publication in a general science journal rather than a more specialist journal.

>>> We are grateful to reviewer #1 for his/her relevant comments and suggestions. We have thoroughly revised the paper and included new genetic analyses to assess the effect of selection on dispersal and competition abilities in *A. thaliana*. In particular, we have now used a multivariate approach for testing selection on correlated traits in *A. thaliana*, based on genetic divergence between populations (Ovaskainen et al., 2011). Moreover, we have now included new analyses of F_{STQ}/F_{ST} and pleiotropy using different SNP subsets. In our answers to specific comments below, we describe in more details the modifications made on the manuscript, providing a more robust genetic basis of our phenotypic trade-offs.

Reviewer #2 (Remarks to the Author):

*This paper addresses an interesting and important question: Can tradeoffs between dispersal, competitive ability, and stress tolerance explain the distribution of *Arabidopsis thaliana* life history phenotypes across Europe? Specifically, can the recent spread of "non-relict" genotypes as human commensals across central Europe be explained by their dispersal ability, at the cost of stress tolerance and competitive ability which were favored at the northern and southern range edges? The study's combination of experimental common garden data with genomic analyses is commendable. The common garden was well designed and executed, although I wish it had been possible to include more accessions. Unfortunately, however, I had a lot of difficulty understanding the methods and have concerns about some of the analyses, especially of reaction norms to stress. Overall, I'm not entirely convinced that the results as presented justify some of the*

conclusions. Nevertheless, if the authors can address the issues below, this study could be a very useful and interesting contribution.

>>> We thank the reviewer for the overall positive remarks on this work and we understand his/her concerns about the lack of clarity in the methodology. Accordingly, we have now clarified several parts of the manuscript, with special attention to the calculation of plant response to competition and water stress. Moreover, we have performed new genetic analyses to reinforce our results. See details below.

Specific comments:

Introduction:

Lines 74 – 78 I was surprised that the authors don't cite relevant experimental evolution papers by Fakheran et al (2010, PNAS) and Williams et al. (2016, Science), which both use the Ler x Cvi A. thaliana RILs (a non-relict x relict cross) to examine evolution of seed mass, competitive ability, and/or dispersal traits. The Fakheran paper in particular addresses competition/colonization tradeoffs and the implications for the current results should be mentioned in the discussion.

>>> We are really grateful to the review for these recommendations. We totally agree about the relevance of these papers for our study, and we have now included them throughout the text, specifically in the introduction (L115-121) and in the discussion section (L320-325 and L351-362).

Line 96. Is the 10,000 year time frame generally accepted? See Fulgione et al. 2018.

>>> This is true that there is certain discrepancy about starting period of *A. thaliana* expansion across Europe. We have now rephrased this part in the introduction (L102-104).

Fig. 1. The dark green region representing the origin of the second expansion is not the same as the origin shown in Figure 5 of Lee et al. 2017.

>>> We thank reviewer for pointing out this error. We have re-arranged figures in this new version and the corrected one can be seen in Figure 1A.

Line 127. There were only 71 accessions in the experiment. Say so here and throughout.

>>> We corrected the number of accessions throughout the text (e.g., L124 in the introduction section, L136 in the results section, L394 in Methods section).

Lines 131-132: objective (iii) is very general and vague and could be sharpened.

>>> We have now rephrased our second and third objectives (L130-132).

Methods:

Lines 335-339: How many accessions were in each of the groups?

>>> The number of accessions per group was included in Supplementary Table 1, but we agree that it is worth communicating these values in the main text (L140-141, L150 in Result section and L399-401 in Methods section).

Line 371: When was maximum reproductive height measured? Were plants harvested and measured at senescence?

>>> Indeed, we measured maximum reproductive height on harvested plants. This is now indicated in the main text (L434-436).

Lines 378- 381., Line 401. I have major concerns about how these ‘individual’ reaction norms were calculated. Reaction norms to course-grained environments like stress and competition are properties of genotypes, not individuals. Consequently, it is inappropriate to run linear models on individual “reaction norms” for these genotypic traits. A simple model run on genotype plasticities calculated from genotypic means in the relevant treatments seems more suitable for comparing tolerances across geographic groups, but power is low. A better option might be to treat fecundity in control, competition, and water stress environments as three separate traits for Q_{st}/F_{st} and examine their multivariate divergence (e.g. Ovaskainen et al. 2011, Genetics). Bivariate plots of control vs treatment fecundities might be quite informative about reaction norm evolution. In any case I would like to see fecundities in competition and water stress presented as traits in in figures 3, 4, S2, S3, and S4. I suspect that may turn out to be interesting tradeoffs in size and fecundity between control and stress treatments that would be worth exploring

>>> We thank the reviewer for pointing out the lack of clarity and for suggesting additional approaches. As suggested, we now present the fecundity values in the three treatments (control, competition and water stress; Fig 1D), as well as bivariate correlations of fecundity between treatments (Supplementary Fig. 3). Following the reviewer’ suggestions, we have now recalculated the fecundity response to competition and water stress using genotypic means instead of individual values. Therefore, fecundity response is estimated through absolute difference in genotypic mean fecundity in treatment (competition / water stress) and control conditions; e.g., L444- L452). Finally, we have now performed the multivariate Q_{ST}/F_{ST} approach from Ovaskainen et al. (2011) to investigate the non-neutral divergence in traits and responses to competition and water stress (L499-509).

Line 391. It’s fine to eliminate outliers, but it would be nice to know whether these were biologically real values, or probable measurement errors.

>>> We have now mentioned it in the text, specifying those ones due to human error in plant identification (confounding plant labels) and those truly measured but spanning 3 sd away around the trait median per accession (Davies & Gather, 1993; Hampel, 1974; Leys et al., 2013; Yang et al., 2023) (L462-465).

Genetic analyses: 71 genomes are quite a small sample size for GWA, especially spanning a large geographic area with population structure. Were any significant peaks detected?

>>> We agree with the reviewer that a sample size of 71 accessions is a low number for performing GWAS (but see, for instance, Aranzana et al., 2005). No significant peaks were detected in our analysis, which it was interpreted as the result of the highly polygenic nature of the growth-related traits associated with dispersal and competition abilities (Kroymann & Mitchell-Olds, 2005; Rockman, 2012; Vasseur et al., 2018). Accordingly, it is supported by our results showing a strong signature of polygenic adaptation using a new analysis based on the method developed by Ovaskainen et al., (2011), which was suggested by the reviewer.

Enrichment of outlier haplotypes: Line 440-441. “genomic proportion of relict ancestry”

might be easier to understand. I had to go back to the Lee paper to be sure I understood what was done.

>>> We thank the reviewer for his/her fair assessment of our work and for his/her suggestion, and we have modified the text accordingly. In addition, we rephrased several parts of the materials and methods (L524) and results (L143-145) sections to better introduce the concept of outlier haplotypes including the window size.

Lines 442-443. Why the top 1%? How sensitive were the results to changes in this cutoff value?

>>> According to reviewer's suggestion, we examined whether F_{STQ}/F_{ST} ratio changes depending on the cutoff value of top-SNPs used to calculate F_{STQ} . We used four cutoff values (0.5%, 1%, 2%, and 5%). F_{STQ}/F_{ST} were not sensitive to variation in the low cutoff values (i.e. 0.5%, 1%, and 2%), but started to show a slightly different pattern at a high cutoff value (5%). With a 5% cutoff value, seed mass was less differentiated among biogeographical groups. The lack of a clear pattern at high cutoff values is expected since we include more genes less, or even not, related to the trait of interest. We decided to keep the 1% cutoff value, which showed the strongest pattern of trait genetic differentiation and it has been also used in other previous studies (Hanemian et al., 2020; Kazakou et al., 2019; Vasseur et al., 2018).

Anyway, we have now included this point in the methods (L516-521) and results (L248-258) sections together with a new figure in supplementary material (new Supplementary Fig. 8).

Results:

Lines 142- 145. This sentence is very confusing – it seems to be the result of a cut and paste error during editing. Also, outlier haplotypes need to be defined and explained briefly here, since most readers of a short format paper like this will read the methods last. The definition of outlier haplotypes in the caption of Fig.1 b is similarly confusing.

>>> As said above, we have now redefined the concept of outlier haplotypes in the text and fig.1A for improving in understanding (L143-145).

Figures 2 and S3. These are correlations, with no assumptions about causality, so regression lines should not be shown. In Fig S3, the figure would be easier to read if r values could be given within the same panels as the scatterplots rather than above the diagonal. Again, the correlations involving water and competition response should be calculated from reaction norms between genotype means in control and treatment, not from individual plants. And fecundities in competition and water stress should also be included as traits in Fig S3. Since fecundity was used to calculate responses to water stress and competition, these are not independent variables, and the observed correlations between them may in part be artifacts of the calculation.

>>> Following the reviewer suggestion, we now presented the correlation coefficients in the same panels as scatterplots (Supplementary Fig. S3). Moreover, the fecundity values in competition and water stress as independent traits have been added and points have been colored according to their geographical group to easier read the figure. But, we kept the correlation line in those panels with significant correlation between variables and we have now clarified that such correlation line does not mean causation between variables in text (L467) to avoid confusion. Moreover, we have notice in the text the lack of independence between fecundity and stress response in fecundity (L451-452).

Lines 188-191. Yes, these correlations are significant, but that's because of the huge sample size. Are they really biologically meaningful evidence of a genetic tradeoff? What would happen if you calculated correlations just with the top 1% of associated SNPs for each trait?

>>> We have now compared the correlations between SNP effects at the whole-genome level and among 1% top-SNPs related to each trait. Negative and positive SNPs correlations were kept using both datasets. In both datasets, the strongest negative correlations were found between SNP effects on fecundity and plant height and SNP effects on response to competition and water stress; and the strongest positive correlation was found between SNP effects related to response to water stress and SNP effects related to response to competition. However, some correlations were found significant at the whole-genome level but were not among 1% top-SNPs (seed mass vs fecundity and plant height, and fecundity vs plant height). We included these new results in the text (L225-234), with a new figure in the supplementary material (Supplementary Fig. 6).

In addition, we also measured the number of SNPs in common among the 1% top-SNPs of different traits. Interestingly, many SNPs were common to different traits, in particular for fecundity related traits. We also included this point in the result section (L244-246), and a new figure in the supplementary material (Supplementary Fig. 7).

Line 196-197. Again, I don't think it is appropriate to calculate Q_{st} for stress response traits based on calculations of "individual" reaction norms. Q_{st} should be reported for fecundities in the three experimental treatments, and the Ovaskainen (2011) method for examining multivariate divergence might be helpful.

>>> As said above, we have now included the analysis from the Ovaskainen et al., (2011) in this new version of the paper. To develop it, we used the new variable of stress response in fecundity calculated from absolute differences in fecundities between treatment using genotypic mean as reviewer suggested.

Line 198 "structuration" should be "structure"

>>> This line was rephrased.

Lines 211-213. Given my concerns about the calculation of these variables, I'm not sure how to interpret these results. I would trust them more if the GWAS to identify top SNPs had been done on reaction norms calculated entirely from genotypic means, as well as separately for fecundity values in all the treatments. Also: why not also examine divergence among non-relict groups?

>>> Our goal was to estimate how genotypes are impacted by drought and competition, which might not be well reflected in the absolute fecundity values under water stress and competition. For instance, a genotype producing a lot of seeds can be the most productive in the different environments, but still be the more impacted (if the impact of stress does not compensate for the larger production of seeds). Thus, using absolute fecundity or relative difference in fecundity will return different results. That is why we decided to use fecundity difference between treatments and control as estimate of genotypic responses to water stress and competition. We added this point in the text (L444-452), and we thank reviewer for these interesting suggestions. According to his/her second comment, we now included a table with pairwise comparison of F_{STQ}/F_{ST} among all groups using 1% top-

SNPs (Supplementary Table 4) and between relict and nonrelict groups for cutoff values of 0.5%, 2% and 5% top-SNPs (Supplementary Table 5).

Discussion:

Lines 239- 240. Although I largely agree with the idea that populations may have differentiated along colonization/competition and colonization/tolerance axes, I think that this sentence overstates what can confidently be concluded from the results presented here.

>>> We rephrased several parts of the discussion to tone down and be clearer about what our results suggest.

One important question that I would like to see a little speculation about is how selection acted on colonization ability. Is this a case of selection for long-distance dispersal ability during the colonization process? Or is it due to selection for dispersal in metapopulations in ruderal disturbed environments? Or was selection on seed size relaxed in ruderal environments, removing constraints on selection response for higher seed number?

>>> We have now introduced some lines explaining potential explanation about how selection acted on colonization ability in the discussion. (L311-325)

Line 242. Fulgione et al. 2018 could also be cited here.

>>> We have now included this reference (L305).

Lines 249- 251. Is there evidence of this tradeoff within regions? Also, how much of the variation among populations can be explained by continuous variation in the environment of origin (e.g. land use type, climate). I really like Figure S5 and it might be nice to put it in place of Fig. 2 (with r and p values shown as in the original and regression line removed).

>>> We have run correlations between seed mass and fecundity across genotypes within each biogeographical group, becoming significant or marginally significant for North nonrelict and relicts groups, respectively ($P < 0.05$). We have included this point in the results section (L173-174) and added a new figure in supplementary material showing the correlation per group (Supplementary Fig. 4).

We have moved Supplementary Fig. 5 to the main text (now as Fig. 2A) according to reviewer's suggestion, we have kept r and p -value but also the regression line explaining in the text that it does not imply causation.

The idea to test variation in key functional traits as function of habitat and other environmental conditions (climate, land use, etc) across Europe is very interesting and could provide supplementary information about our phenotypic patterns. However, we consider that such idea deserves of thorough investigation of the potential environmental variables that trigger functional differences across geographic groups. While very valuable, this is clearly out of the scope of the present study.

Lines 268- 269. I think this statement is an overreach given the tiny genomic correlation coefficients observed. Although significant due to the large sample size, they don't

provide strong evidence of pleiotropy or genetic constraint. The authors might also consider the possibility that the observed phenotypic syndromes emerged through correlated multigenic selection on entirely independent loci underlying the different traits. How much overlap was there in the top SNPs for the different traits?

>>> We thank the reviewer for raising this interesting point. As suggested, we estimated the number of SNPs in common among the 1% top-SNPs (4,748 SNPs) identified for each trait. Consistent with the idea of pleiotropy between our traits, we found that substantial number of top-SNPs were shared between traits, in particular between fecundity-related traits (see comment above). We included a new figure in supplementary information showing the number of 1% top-SNPs overlapping between traits (Supplementary Fig. 7) and discussed this point in the text (L245-246).

Lines 272- 273. Fakheran et al. (2010) and probably Williams et al (2016) could also be referenced here.

>>> We have enriched the analysis of our results mentioning these two studies in the discussion (L320-325; L352-356).

Figure S2. This figure would be more useful if it could be arranged according to geographic groups, or at least color-coding by group. Also please include fecundity in competition and stress treatments as traits.

>>> Following the reviewer's recommendation, we have modified the figure reordering boxplots according to geographic groups. Moreover, this figure now includes the graphical representation of fecundity in competition and water stress (Supplementary Fig. 2).

Reviewer #3

*In this study the authors find evidence for SNPs associated with phenotypic trade-offs between colonization and competitiveness traits. These SNPs vary in frequency across *A. thaliana* range in a manner that might support the expectations for trade-off roles in species expansion, which has been rarely demonstrated empirically. The approach taken is novel and should be of interest to the evolution and ecology readership of the journal. However, I found very confusing to follow the narrative of the due to crossover of terms that do not necessarily means the same thing. Thus, I am not sure the conclusion are fair given the actual results, but possibly they are. There are two main issues that need clarification:*

>>> We thank reviewer #3 for his/her valuable comments. We thoroughly revised the manuscript to clarify the hypotheses and the method used (see also our responses to comments from other reviewers above).

1) *The geographical groups used and their meaning. "Central" versus "marginal" populations are the concepts used in the introduction, which implies that the relevance of trade off in limiting population expansion is going to be considered. However, the accessions sampled are from north, south and central Europe, and accessions are divided into "relict" and "non-relict". Is not clear how significant difference between the southern relict group from other groups actually translate to central or marginal populations and relate to theoretical expectations. Chuang & Peterson (2016) review indicate that theoretical expectations are of shifts toward greater reproductive output at some edge demes, which may aid in the establishment of a species in a novel environment. I don't think that is what was found here?*

>>> We thank the reviewer for this interesting point and included a discussion on it in the revised paper (L307-325).

*In the abstract it is said “We showed that a trade-off between plant fecundity and seed mass constrains the diversification of *A. thaliana* in Europe”. But did the experiment tested directly whether relict genotypes can outcompete the cosmopolitan genotype? And is this equivalent to saying that there are differences between peripheral and central genotypes as stated in the abstract?*

>>> No, this was not tested in the present study, where we focused on absolute variability on competitive ability and stress tolerance, not relative variation between genotypes. Given the type of reproduction of *A. thaliana*, competition is likely to be predominantly driven by competition between conspecifics (from seeds originated from the same mother plant), but we agree that a promising direction to take in the future is to test the competitive advantage of relicts over nonrelicts. To avoid confusion, we have rephrased this sentence in the abstract (now L36). Moreover, we have rephrased the introduction to better explain this point and added a new paragraph (L50-66 in the introduction and L286-305 in the discussion).

There was clear demonstration of trade-offs across accessions, and that it is likely this trade off has a genetic basis (measurements of heritability and genetic covariance would be helpful here), but how does that show that the trade-off constrained diversification? It would be helpful if the connections between theoretical expectations and the geographical groups connected are more clearly shown.

>>> We recognized that the word ‘diversification’ was not used correctly in our study, that is why we have now avoided to use this word. We agree that a better explanation of the evolutionary implications of our findings needed to be included. Accordingly, we rephrased the introduction and the discussion. We also included measurement of heritabilities (L181-186 in results; L519-521 in methods and Supplementary Table 3).

2) *Population genetic models predictions and estimates are described as fait accompli. Much of the experimental design is relying on assumptions about the relationship between the geographical groups and how this relates to models of potential expansion of *Arabidopsis*; however these assumption need to be made more clearly. For example, I assume that the south to north choice of populations is based on the assumption that this is how *Arabidopsis* recolonized Europe after surviving in refuges in the Iberic peninsula. This would suggest though that the northern group is the “peripheral” or “marginal” group; but then the relicts and non- relicts comes on top of that, and it becomes quite difficult to match expectations and model estimates. This is further confused as the definition of relicts and their relevance is not entirely clear. But perhaps more importantly, is good to remember George Box’s famous quote: “All models are wrong, but some are useful”. There is indeed a good probability that recolonizations did happened as models predict... however, they are not “known” or “demonstrated”. The colonizations happened millions of years ago and we can only speculate that this is what happens. Thus, I suggest the language around much of the introduction and the discussion is toned down to reflect that all the population models for *Arabidopsis* colonization scenarios are not facts. For example is plausible that new lineages had to compete with older lineages, but it is also possible that there were new environments that new lineages moved in without no need to compete with any lineage. None of this invalidate the findings of the manuscript, which are interesting. It just should be presented in a more open minded way, i.e. if this is about range expansion, limitations of adaptations at the*

margins, there should be more objective ways to define “margins” than grouping into relics and cosmopolitan genotypes.

>>> We thank the reviewer for point out the lack in clarity about the demographic history of *A. thaliana* in Europe, as well as for the need to tone down some statements. According to his/her suggestions, and as said above, we thoroughly rephrased the introduction and the discussion, notably to highlight the uncertainties in our knowledge of the evolutionary history of *A. thaliana* in Europe. Note however that the history of European recolonization by *A. thaliana* is relatively recent (only 50-10 ka, not millions of years) and has been carefully examined thanks to the massive amount of genetic information available in this species. We hope this new version better presents the rationale of the study and how it is built on previous genetic findings that shed lights on the evolutionary history of *A. thaliana* in Europe (e.g., L93-121 in the introduction and L302-329 in the discussion).

There is overall an issue with clarity of approach and methods, and here is a non-exhaustive list of issues that could be clarified for the methodology and results to be better understood:

- *How was the competition treatment established? Were seedlings transplanted into a pot? If so at what age? As written it seems like they were transplanted after coming out of the cold? That would be quite late in their life-history.*

>>> We appreciate the reviewer’s concerns. We agree with reviewer that the methodology about how the competition treatment was established was not expressed very clearly. Seedlings were not transplanted but seeds were sowed in competition. We planted 3-4 seeds per pot and position (center and four corners), keeping the first germinated plant per position and removing the rest. We have now clarified it in the text (L410-415).

- *Fecundity is calculated in centimeters? It is described as the multiplication of fruit number (including all branches, or just in the main branch? The latter could introduce many biases) by fruit length, but there is no unit associated with the variable in graphs and tables*

>>> We have now specified that we counted the fruit number at whole-plant level. Also, we have now clarified that our fecundity index is a proxy of seed number per plant, measured as number of fruits x average fruit length, following the same methodology than previous studies (e.g., Wang et al., 2022) (L440-441)

- *What is meant by the “reaction norm” (line 379)? Do you mean the slope of a regression is calculated? Is confusing, as it also says “difference” in fecundity of a particular individual? If is just the difference as in the subtraction of fecundity from the average fecundity in the control treatment, the term “norm of reaction” is not necessary or accurate. Furthermore, absolute differences will bias the estimate of competition towards accessions that produces more seeds, as the maximum competition would be constrained by the maximum fecundity. Perhaps using slopes would be better to remove such biases*

>>> We agree with the reviewer that the description of variables used to measure stress response in fecundity lacked clarity in the previous version of the manuscript. We modified the text to solve this issue. In this study, we have used the difference in genotypic mean fecundity between treatment and control to quantify plant response to

water stress and competition. We clarified this point and avoid using the term “reaction norms”, which might be confusing (see comments above). As pointed by the reviewer, this measurement is necessarily influenced by the number of seed produced in control condition. However, we believe here that, in an evolutionary perspective, this is the absolute “amount of progeny lost” what is relevant for population demography. Actually, there is an active debate in the literature about the use of relative versus absolute fitness, with no clear consensus (e.g., Jin & Tsang, 2005; Orr, 2007, 2009; Reiss, 2013; Wilson, 2004). As said above, our goal was to estimate how drought and competition might impact the demographic success of a given genotype, which is more accurately measured by changes in absolute fecundity rather than relative fecundity (*i.e.* standardized by fecundity in control). For instance, a genotype producing a lot of seeds can be the most productive in the different environments, but still be the more impacted (if the impact of stress does not compensate for the larger production of seeds). Using relative differences in fecundity will return interesting results related to the efficiency of a given genotype to deal with stress and/or competition, but which is less relevant for assessing its fitness and demography, and the effect of environmental factors on these parameters. We added this point in the text (L444-452).

- *There is no definition of “outlier haplotypes” or how they were determined on the methods section before line 439; and no clarification on that section either. On the results section, there is a definition (line 143): “alleles less frequent across whole species genome”, but without knowing what is an allele (a single SNP, or a haplotype of a certain window size?) this terms remains unclear.*

>>> We have now more clearly defined the “outlier haplotype” in the result section (L143-145) and Materials and Methods (L524)

- *Line 407 – I assume SNPs with frequencies higher than 5% were kept? Your sentence suggests the opposite*

>>> Indeed, we used minor allele frequency (MAF) filter to keep only SNPs with MAF > 5%. Included in the Material and Methods (L480).

- *Line 419 – I am unclear whether I understood this sentence. It is unclear what the “pair” is in “all pairwise combinations of SNP effects on the phenotypic traits”?*

>>> We have reworded it for greater clarity(L490-491).

- *Line 422, results and discussion – The comparison of Q_{st} to F_{st} provides insight on whether there is departure from a neutral model. This may be caused by adaptation, but it can also be due to other causes, thus the use of the term “adaptive should be used with more caution.*

>>> We have reworded this sentence accordingly (L493)

- *NASC stands for Nottingham Arabidopsis stock centre (line 341)*

>>> Changed (now L404) and in table 1.

References

- Aranzana, M. J., Kim, S., Zhao, K., Bakker, E., Horton, M., Jakob, K., Lister, C., Molitor, J., Shindo, C., Tang, C., Toomajian, C., Traw, B., Zheng, H., Bergelson, J., Dean, C., Marjoram, P., & Nordborg, M. (2005). Genome-Wide Association Mapping in Arabidopsis Identifies Previously Known Flowering Time and Pathogen Resistance Genes. *PLOS Genetics*, *1*(5), e60. <https://doi.org/10.1371/journal.pgen.0010060>
- Davies, L., & Gather, U. (1993). The Identification of Multiple Outliers. *Journal of the American Statistical Association*, *88*(423), 782–792. <https://doi.org/10.1080/01621459.1993.10476339>
- Hampel, F. R. (1974). The Influence Curve and its Role in Robust Estimation. *Journal of the American Statistical Association*, *69*(346), 383–393. <https://doi.org/10.1080/01621459.1974.10482962>
- Hanemian, M., Vasseur, F., Marchadier, E., Gilbault, E., Bresson, J., Gy, I., Violle, C., & Loudet, O. (2020). Natural variation at FLM splicing has pleiotropic effects modulating ecological strategies in Arabidopsis thaliana. *Nature Communications*, *11*(1), Article 1. <https://doi.org/10.1038/s41467-020-17896-w>
- Jin, N., & Tsang, E. (2005). Relative fitness and absolute fitness for co-evolutionary systems. *Proceedings of the 8th European Conference on Genetic Programming*, 331–340. https://doi.org/10.1007/978-3-540-31989-4_30
- Kazakou, E., Vasseur, F., Sartori, K., Baron, E., Rowe, N., & Vile, D. (2019). Secondary metabolites have more influence than morphophysiological traits on litter decomposability across genotypes of Arabidopsis thaliana. *New Phytologist*, *224*(4), 1532–1543. <https://doi.org/10.1111/nph.15983>
- Kroymann, J., & Mitchell-Olds, T. (2005). Epistasis and balanced polymorphism influencing complex trait variation. *Nature*, *435*(7038), 95–98. <https://doi.org/10.1038/nature03480>
- Leys, C., Ley, C., Klein, O., Bernard, P., & Licata, L. (2013). Detecting outliers: Do not use standard deviation around the mean, use absolute deviation around the median. *Journal of Experimental Social Psychology*, *49*(4), 764–766. <https://doi.org/10.1016/j.jesp.2013.03.013>
- Orr, H. A. (2007). Absolute Fitness, Relative Fitness, and Utility. *Evolution*, *61*(12), 2997–3000. <https://doi.org/10.1111/j.1558-5646.2007.00237.x>
- Orr, H. A. (2009). Fitness and its role in evolutionary genetics. *Nature Reviews Genetics*, *10*(8), Article 8. <https://doi.org/10.1038/nrg2603>
- Ovaskainen, O., Karhunen, M., Zheng, C., Arias, J. M. C., & Merilä, J. (2011). A New Method to Uncover Signatures of Divergent and Stabilizing Selection in Quantitative Traits. *Genetics*, *189*(2), 621–632. <https://doi.org/10.1534/genetics.111.129387>
- Reiss, J. O. (2013). Does selection intensity increase when populations decrease? Absolute fitness, relative fitness, and the opportunity for selection. *Evolutionary Ecology*, *27*(3), 477–488. <https://doi.org/10.1007/s10682-012-9618-7>
- Rockman, M. V. (2012). The QTN program and the alleles that matter for evolution: All that's gold does not glitter. *Evolution; International Journal of Organic Evolution*, *66*(1), 1–17. <https://doi.org/10.1111/j.1558-5646.2011.01486.x>
- Vasseur, F., Exposito-Alonso, M., Ayala-Garay, O. J., Wang, G., Enquist, B. J., Vile, D., Violle, C., & Weigel, D. (2018). Adaptive diversification of growth allometry in the plant Arabidopsis thaliana. *Proceedings of the National Academy of Sciences*, *115*(13), 3416–3421. <https://doi.org/10.1073/pnas.1709141115>

- Wang, P., Meng, F., Donaldson, P., Horan, S., Panchy, N. L., Vischulis, E., Winship, E., Conner, J. K., Krysan, P. J., Shiu, S.-H., & Lehti-Shiu, M. D. (2022). High-throughput measurement of plant fitness traits with an object detection method using Faster R-CNN. *The New Phytologist*, *234*(4), 1521–1533. <https://doi.org/10.1111/nph.18056>
- Wilson, D. S. (2004). What is wrong with absolute individual fitness? *Trends in Ecology & Evolution*, *19*(5), 245–248. <https://doi.org/10.1016/j.tree.2004.02.008>
- Yang, R., Liu, H., & Li, Y. (2023). Quantifying uncertainty of marine water quality forecasts for environmental management using a dynamic multi-factor analysis and multi-resolution ensemble approach. *Chemosphere*, *331*, 138831. <https://doi.org/10.1016/j.chemosphere.2023.138831>

REVIEWER COMMENTS

Reviewer #2 (Remarks to the Author):

The authors have put a lot of effort into responding to the reviews, and have done a good job responding to most of my comments (with one minor exception I will note below) The addition of the new analysis reported in figure 2B is particularly compelling: they have shown convincing evidence of adaptive differentiation along the axis of the tradeoff between seed mass and fecundity. I noticed a few minor remaining issues but these are easy to address. The paper does need another round of copyediting to correct a few minor typos and errors in the text- but overall this is a great improvement on the first version.

Here some comments and minor suggestions:

Lines 67- 69: This new sentence is confusing. How about something like "With a limited resource budget, increased allocation to one strategy leads to reduced allocation to the other strategy, resulting in a trade-off". Or just delete the sentence.

Line 173, Figures 2A and S3: I still don't see the point of showing "correlation lines" in these figures. It's not standard practice to show regression lines when there is no causal relationship and the reported statistics are correlation coefficients. The only exception might be if the authors were interested in the allometry of these relationships and reporting slopes from type II regressions. But that's not the case here, and I still recommend removing the lines. Aside from this, the figures are greatly improved in this version.

Lines 177-179: I appreciate that the authors added a caveat about non-independence of these variables in the methods section, but it might be a good idea to put that here too.

Line 190: How was Q_{st} calculated for response of fecundity to competition and drought calculated from family means? I don't see where the variances came from.

Lines 220-222: With the low number of lines, power to detect associations was also limited

Lines 243-245: There was less power to detect significant correlations with only the top 1% SNPs, but interesting the magnitude of the correlation coefficients was higher for the top SNPs than genome wide, which may be worth mentioning.

Line 314: "poorer competitors but better dispersers"- in other words something like Grime's ruderal strategy. In general, this paragraph of the discussion is well done and helps to put the results in a larger context- a great improvement over the last version.

Lines 424-426: What was the daylength regime? Ambient seasonal photoperiod? Or if it was supplemented, what was the light/dark cycle?

Lines 438: Sentence needs to be fixed to correct subject-verb disagreement. E.g. "We harvested plants when they reached maturity..... and then performed a set of phenotypic measurements" Or "Plants were harvested.... andphenotypic measurements were performed"

Line 453: "what"-> "that"

Line 470: I still don't understand what the authors mean by "linear correlations". I assume that they mean genotypic mean correlations.

Lines 476- 478. Please be clear about the model. Same fixed structure, but run on genotype means?

Fig 2 B: This figure is really a great addition, but please explain in the legend what the different ovals represent

Reviewer #3 (Remarks to the Author):

First, I would like to acknowledge the huge amount of work the authors have done into responding to the previous reviewer's comments. The manuscript is much improved, and the description of what is done is clearer. However, while the aims of the study and the approach has been clarified the connection between what was done, and the broad questions being asked, are still not clearly laid out. The main concern here is the choice of accessions (100masl is not very common...) and the allocation of them into the 5 "biogeographical groups". The interpretation of the results very much depend on the rationale for these groups.

The match between what is written as objectives and the use of the 5 biogeographical groups is unclear. Obj 1 and 2 talks about "spatial structure" and the "dynamics of wild populations"; but all the results are focused on comparisons of these 5 biogeographical groups that are not clearly spatially separated. Figure 1A shows North non-relict accessions located deeply within the central region, or perhaps 3 central ones quite mixed with northern ones (depending on definition of what is central or north). Equally, there is a central accession quite far south. To address the objective 1 as written I would think it would be necessary to first investigate the spatial structure of the accession and group them by environmental, latitude/longitude, or genetic similarity. Alternatively, objectives 1 and 2 needs to be rewritten to justify the 5 somewhat arbitrary groups created. The confusion is enhanced by the fact that subtitle for describing the first results says "center to margin differentiation of *A. thaliana* populations", but your statistical test uses 5 groups where marginality is not very well defined. By line 166, the comparison is defined in terms of "historical demography" (which is fine), but then only compares north and south relicts to central non-relicts. I wonder if the rationale for the 5 groups was to do a longitudinal study, and marginal are the south and north AND a historical study? If so, the two variables should be looked separately, or as a multivariable problem, because the interpretation of the 5 "biogeographical units" is shaky, and is not fitting either narrative.

The fact that you are observing statistically significant differences among the 5 groups is reassuring towards the idea that there are some real biological phenomena. However, the sample sizes for some groups are quite small, and others very big. Thus, if you would keep at 5 regions due to some clearer explanation, I would be more reassured if lines were randomly assigned to groups of the same sizes as the current five groups and re-analyzed (as a bootstrap) to determine the significance of the difference between the groups.

Line 220 – The lack of significant SNPs is most likely due to the lack of power, it says nothing about the genetic architecture of these traits (even though I would agree that these traits are likely to be polygenic). This sentence should be deleted. Equally, given the experimental design it is not possible to say anything about the molecular basis of these traits (again, no power); so the idea that this is being done for identifying the genetic basis of these traits should be removed from the study (especially from objective 2).

I remain unclear about two technical issues:

- 1) Fruits were counted when EACH plant senesced, or did you have an end point when the fecundity data was collected for ALL plants? The concern here is that there is an intrinsic trade of between vegetative size and seed production, determined by developmental time. With a fixed cut-off, plants that flower later, would have at that point bigger vegetative and small reproductive allocation relative to early flowering plants.
- 2) I understand that information to classify accessions into "relicts" comes from Lee et al study. However, Line 141 and figure 1 B is presenting data on the number of haplotype outlier, as a result of this study. If it is a new analysis, details of it needs to be included in the M&M. If not, this should be removed from the results section.

Last, but not least, I think there is a general over interpretation of the results. This is clearly illustrated by the statement on line 184, which is overly strong. The results are in fact "compatible" with the explanation. Since the experiment was done under lab conditions, the statement that "these results confirm that phenotypic diversity in *A. thaliana* is SHAPED by A MAJOR trade-off between maximizing fecundity and dispersal (numerous light seeds and tall inflorescences) and maximizing competitive ability and stress tolerance." cannot be concluded. I recommend toning down the significance of the results in the discussion also throughout.

RESPONSE TO REVIEWERS COMMENTS

Reviewer #2 (Remarks to the Author):

The authors have put a lot of effort into responding to the reviews, and have done a good job responding to most of my comments (with one minor exception I will note below) The addition of the new analysis reported in figure 2B is particularly compelling: they have shown convincing evidence of adaptive differentiation along the axis of the tradeoff between seed mass and fecundity. I noticed a few minor remaining issues but these are easy to address. The paper does need another round of copyediting to correct a few minor typos and errors in the text- but overall this is a great improvement on the first version.

>>> We thank the reviewer for the overall positive remarks on this work. We have now clarified all minor reviewer concerns. See details below.

Here some comments and minor suggestions:

Lines 67- 69: This new sentence is confusing. How about something like “With a limited resource budget, increased allocation to one strategy leads to reduced allocation to the other strategy, resulting in a trade-off”. Or just delete the sentence.

>>> We have clarified the sentence (L67-68).

Line 173, Figures 2A and S3: I still don't see the point of showing “correlation lines” in these figures. It's not standard practice to show regression lines when there is no causal relationship and the reported statistics are correlation coefficients. The only exception might be if the authors were interested in the allometry of these relationships and reporting slopes from type II regressions. But that's not the case here, and I still recommend removing the lines. Aside from this, the figures are greatly improved in this version.

>>> As suggested, we have now deleted the correlation lines from both figures.

Lines 177-179: I appreciate that the authors added a caveat about non-independence of these variables in the methods section, but it might be a good idea to put that here too.

>>> Added (L202-204).

Line 190: How was Q_{st} calculated for response of fecundity to competition and drought calculated from family means? I don't see where the variances came from.

>>> Clarified in the text (L547).

Lines 220-222: With the low number of lines, power to detect associations was also limited

>>> We appreciated the reviewer concern and we have included it in the text (L246-247).

Lines 243-245: There was less power to detect significant correlations with only the top 1% SNPs, but interesting the magnitude of the correlation coefficients was higher for the top SNPs than genome wide, which may be worth mentioning.

>>> We have added this interesting remark in the text (L269-270).

Line 314: “poorer competitors but better dispersers”- in other words something like Grime’s ruderal strategy. In general, this paragraph of the discussion is well done and helps to put the results in a larger context- a great improvement over the last version.

>>> We thank the reviewer for the positive comments on this updated version. We included this remark about Grime’s strategy space in the discussion (L340-341).

Lines 424-426: What was the daylength regime? Ambient seasonal photoperiod? Or if it was supplemented, what was the light/dark cycle?

>>> There was not a supplemented light in the greenhouse experiment, but plants germinated and grew under ambient photoperiod. We have clarified it in the text (L466-467).

Lines 438: Sentence needs to be fixed to correct subject-verb disagreement. E.g. “We harvested plants when they reached maturity..... and then performed a set of phenotypic measurements” Or “Plants were harvested.... andphenotypic measurements were performed”

>>> Thank you for the appreciation. We have corrected it.

Line 453: “what”-> “that”

>>>Changed

Line 470: I still don’t understand what the authors mean by “linear correlations”. I assume that they mean genotypic mean correlations.

>>> Changed.

Lines 476- 478. Please be clear about the model. Same fixed structure, but run on genotype means?

>>> Thank you for this concern, we have now specified it in the text (L510-L517).

Fig 2 B: This figure is really a great addition, but please explain in the legend what the different ovals represent

>>> We have now included a brief explanation about the meaning of ellipses in the legend of the figure.

Reviewer #3 (Remarks to the Author):

First, I would like to acknowledge the huge amount of work the authors have done into responding to the previous reviewer's comments. The manuscript is much improved, and the description of what is done is clearer. However, while the aims of the study and the approach has been clarified the connection between what was done, and the broad questions being asked, are still not clearly laid out. The main concern here is the choice of accessions (100masl is not very common...) and the allocation of them into the 5 "biogeographical groups". The interpretation of the results very much depend on the rationale for these groups.

We are grateful to reviewer #3 for his/her relevant comments and suggestions. We have revised the paper and included a more compelling explanation about the accession classification. We have also included a new analysis comparing observed phenotypic patterns with those generated by random permutations on biogeographical groups to detect chance in our phenotypic pattern. Moreover, we provide new analyses of trait-latitude relationships. See details below.

*The match between what is written as objectives and the use of the 5 biogeographical groups is unclear. Obj 1 and 2 talks about "spatial structure" and the "dynamics of wild populations"; but all the results are focused on comparisons of these 5 biogeographical groups that are not clearly spatially separated. Figure 1A shows North non-relict accessions located deeply within the central region, or perhaps 3 central ones quite mixed with northern ones (depending on definition of what is central or north). Equally, there is a central accession quite far south. To address the objective 1 as written I would think it would be necessary to first investigate the spatial structure of the accession and group them by environmental, latitude/longitude, or genetic similarity. Alternatively, objectives 1 and 2 needs to be rewritten to justify the 5 somewhat arbitrary groups created. The confusion is enhanced by the fact that subtitle for describing the first results says "center to margin differentiation of *A. thaliana* populations", but your statistical test uses 5 groups where marginality is not very well defined. By line 166, the comparison is defined in terms of "historical demography" (which is fine), but then only compares north and south relicts to central non-relicts. I wonder if the rationale for the 5 groups was to do a longitudinal study, and marginal are the south and north AND a historical study? If so, the two variables should be looked separately, or as a multivariable problem, because the interpretation of the 5 "biogeographical units" is shaky, and is not fitting either narrative.*

>>> We agree with reviewer that the criteria for accession classification was not enough clear in the last version. The accessions were classified considering two criteria: geographical position (south, central, north) and genetic group (origin). Indeed, previous studies in *Arabidopsis* demonstrated that, in Europe, the species can be subdivided into nine genetic groups¹, including relict vs nonrelict. These groups structure the species diversity and reflect the historical demography of *Arabidopsis* (with non-relicts

expanding in all directions few thousands years ago). Thus, the distribution of *Arabidopsis* accessions results from the strong interplay between genetic origin and geographical position. In our study, relict vs non-relict classification was prioritized since we expect phenotypic patterns related to dispersal and competition to be strongly related to the demographic history of the species. Accordingly, for most accessions in our study, there was a congruent classification in genetic and geographical groups (i.e. relicts in the south, close-to-relict in the north), but for a small number of accessions (5 out of 71), there was a mismatch between geographical position and genetic group. These five accessions currently occupy south or north areas while they are genetically similar to those accessions located at central European areas, i.e. they belong to ‘Germany’, ‘Central Europe’ or ‘Western Europe’ genetic group. This is why in our study they are classified within the ‘central nonrelict’ group despite their geographical position. As we already introduced in our study, central accessions quickly dispersed to north and south areas. Thus, central accessions are likely currently expanding (and potentially hybridizing) northward and southward, which might explain why our classification outlined some central accessions outside of their core geographical niche.

We have now included a brief mention of it in the introduction (L106-107 – L111-112). We have expanded results (L138-168) and M&M (L426-435) sections with an explanation about accession classification in detail. Moreover, in supplementary Table S1, we have included the genetic group at which each study accession belong to, according to the 1001 Genomes Consortium study¹ and, we have also represented the phenotypic traits for genetic cluster origin of study accessions (new Supplementary Fig. 4). Using latitude and genetic cluster as x axis, we observed similar phenotypic variation patterns to those observed using our accession classification (mixed geographical and genetic origin).

¹The 1001 Genomes Consortium. 1,135 Genomes reveal the global pattern of polymorphism in *Arabidopsis thaliana*. *Cell* **166**, 492–505 (2016).

The fact that you are observing statistically significant differences among the 5 groups is reassuring towards the idea that there are some real biological phenomena. However, the sample sizes for some groups are quite small, and others very big. Thus, if you would keep at 5 regions due to some clearer explanation, I would be more reassured if lines were randomly assigned to groups of the same sizes as the current five groups and re-analyzed (as a bootstrap) to determine the significance of the difference between the groups.

>>> Following the reviewer suggestion, we have now included the comparisons of null vs. observed values for the mean phenotypic trait per each geographical group. Null models have been generated by randomly shuffling labels (100 replicates) on geographical group for each accession, while keeping sample sizes fixed for each geographical group. The results showed that south relict and North close-to-relict groups have, on average, bigger seeds, greater response to competition and smaller fecundity and plant height values than would be expected by chance. The opposite pattern was observed for center nonrelict group.

We have now included these results in the text (L184-188), but also we explain the model in material and methods section (L520-525) and finally, we have added a new figure showing the results (new Supplementary Fig. 5).

Line 220 – The lack of significant SNPs is most likely due to the lack of power, it says nothing about the genetic architecture of these traits (even though I would agree that these traits are likely to be polygenic). This sentence should be deleted.

>>> We understand the reviewer concern about the lack of power due to the limited number of natural lines (n = 71), also raised by the reviewer 2. Following the indication, we have removed explanation about polygenic nature of these traits and included the lack of power as potential explanation to our lack of correlation (L246-247).

Equally, given the experimental design it is not possible to say anything about the molecular basis of these traits (again, no power); so the idea that this is being done for identifying the genetic basis of these traits should be removed from the study (especially from objective 2).

>>> We have reformulated more carefully the objective 2 (L132-133).

*I remain unclear about two technical issues:
1) Fruits were counted when EACH plant senesced, or did you have an end point when the fecundity data was collected for ALL plants? The concern here is that there is an intrinsic trade off between vegetative size and seed production, determined by developmental time. With a fixed cut-off, plants that flower later, would have at that point bigger vegetative and small reproductive allocation relative to early flowering plants.*

>>> We conducted plant-specific harvests and fruit counts at the point of maturity of each plant to avoid bias in allocation resources in vegetative or reproductive organs between in late and early flowering plants. We have now explained it in material and methods section (L476-478).

2) I understand that information to classify accessions into “relicts” comes from Lee et al study. However, Line 141 and figure 1 B is presenting data on the number of haplotype outlier, as a result of this study. If it is a new analysis, details of it needs to be included in the M&M. If not, this should be removed from the results section.

>>> The information to classify accessions into different genetic groups, including their relict or nonrelict origin, comes from the 1011 Genomes Consortium¹. From Lee et al, we have only used the variable that quantifies the number of haplotype outliers for each accession across 1,135 Genomes. While Lee et al studied the variation of the number of haplotypes against latitude across genome; we used this variable with different goals (i) to support our accession classification (classification based on geographic and genetic similarity, above explained) and (ii) to study how many haplotype outliers are strongly related to our phenotypic traits.

We have now included a brief explanation of this analysis in M&M (L441-443) and we have rewritten the result section (L151-153) to distinguish our results from Lee et al.

¹The 1001 Genomes Consortium. 1,135 Genomes reveal the global pattern of polymorphism in *Arabidopsis thaliana*. *Cell* **166**, 492–505 (2016).

Last, but not least, I think there is a general over interpretation of the results. This is clearly illustrated by the statement on line 184, which is overly strong. The results are in fact “compatible” with the explanation. Since the experiment was done under lab conditions, the statement that “these results confirm that phenotypic diversity in A. thaliana is SHAPED by A MAJOR trade-off between maximizing fecundity and dispersal (numerous light seeds and tall inflorescences) and maximizing competitive ability and stress tolerance.” cannot be concluded. I recommend toning down the significance of the results in the discussion also throughout.

We have tone down this sentence (L196-198) together with others in the discussion (L341; L344).

REVIEWER COMMENTS

Reviewer #2 (Remarks to the Author):

The authors have responded appropriately to the major issues raised in the last round of reviews, and the resulting manuscript will be of interest to a broad audience of ecologists and evolutionary biologists. One more optional suggestion: if the names of the groups in Fig 2B could be replaced with smaller symbols, their positions would be clearer on the figure.

My only remaining concern is that the paper needs a round of careful copyediting for minor typos and English usage. I note a few examples below but this is by no means a comprehensive list.

L 106: Confusing sentence. How about "Most of the European nonrelict accessions are classified within eight genetic clusters".

L 178: "grew" -> "grown"

L 250: $r = 0.01$ for seed mass is not a negative correlation- is the minus sign missing here?

L 382 "perturbated" -> "perturbed"

Methods should be written consistently in the past tense, e.g.

L 435 "select" -> "selected"

L 510, 516, 521 "run" -> "ran"

L 547 "compare" -> "compared"

Missing articles:

L 431: should be "a few thousand years ago"

L 435: "the South relict genetic cluster..."

L. 452 word missing- should be "under well watered conditions"

L 529: "sequenced" -> sequences

Reviewer #3 (Remarks to the Author):

I appreciate the clarification into the classification of the accessions. However, I remain unclear about the rationale behind the groups used. The choice of north, central and south, suggest the importance of latitudinal variation, but investigation of its existence is not discussed. Most of the study is about comparing relicts to non-relicts; but the biggest significant differences are between central relicts, and north/south non-relicts. It is not clear why this should be the most interesting comparison, or the one expected. Any differences that are SOLELY due to relict vs. non-relict status, should be most directly tested by comparing south relict to non-relict. Any statistically significant difference among all 5 groups only indicates that there is variation among groups.

This study clearly shows that central non-relict accessions are quite different from south and north relicts, but given that this difference is not replicated in comparisons of relict and non-relict accessions in the north and south, I don't think is possible to assign this difference unambiguously to the fact that central accessions are non-relicts (why do the authors think that differences between relicts and non-relict change in direction in the north and south for most traits?). Hence, the request for an overall toning down of the conclusions (not only in the example from the last review).

The current editing of the manuscript has unfortunately not solved the mismatch between aims, experimental design and conclusions:

Aim 1: to assess the spatial structure of phenotypic variation in key traits related to dispersal, competition and stress-response abilities in A. thaliana, This aim is very unclear. What is the null expectation for such aim? No differences between all 5 regions? Would one significant difference be sufficient to say there is "spatial structure"? If there is one region different what would this tell us about big patterns of trade-offs?

Aim 2 : to test whether competition-colonization trade-offs determine the distribution of A. thaliana lines, The methodology of this study only allows inferences about correlational patterns, not whether the trade-off determined the distribution of A. thaliana. This would require a phylogenetic approach.

With regards to this aim is also worth pointing out that the R values supporting the idea of a genetic trade-off (figure 9A) are quite small (except for the negative trade off between fecundity under control and competition). All R values associated with seed mass is <0.01!

Aim 3: to explain the demographic success of the nonrelict lineages in Europe.

I cannot see how any of the data collected can "explain the demographic success", but I believe what is being hypothesized here is the success of non-relicts is associated with increase in competitiveness at the cost of fecundity. If that is the case, the aim should be rewritten to clearly state what is being tested, and the conclusion should be that this is likely, but not unambiguous.

Other points:

Given that none of the aims is focused on "central" or "margins", the first subtitle of the results should be changed to reflect aim 1; or aim 1 needs to be changed to reflect this first sub-title. In this section variation in phenotype between relict and non-relict populations seem to be the main focus.

Central non-relicts response to water stress is NOT significantly different to the other groups (Figure 1), and line 183 should be modified to reflect this.

There is good evidence that seed size has been under selection (Figure 2), but very weak indication that plant height (key for dispersal, according with the manuscript) has been under selection. Interestingly, genetic differentiation for fecundity between southern relicts and central non-relicts are also not significant. Thus, again the results are very interesting, but does not unambiguously suggest that relicts were selected for dispersal.

Sentence added on line 106 of the word document does not make sense.

Line 232, the data being discussed in here is shown where? No figures referred....

RESPONSE TO REVIEWERS' COMMENTS

Reviewer #2 (Remarks to the Author):

The authors have responded appropriately to the major issues raised in the last round of reviews, and the resulting manuscript will be of interest to a broad audience of ecologists and evolutionary biologists. One more optional suggestion: if the names of the groups in Fig 2B could be replaced with smaller symbols, their positions would be clearer on the figure.

My only remaining concern is that the paper needs a round of careful copyediting for minor typos and English usage. I note a few examples below but this is by no means a comprehensive list.

>>> We thank the reviewer for pointing out the need for English usage. The text has been carefully corrected. We also modified the Fig 2B, including a legend with the color of lines assigned to each accession group.

L 106: Confusing sentence. How about “Most of the European nonrelict accessions are classified within eight genetic clusters”.

>>> Modified.

L 178: “grew” -> “grown”

>>> Modified.

L 250: $r = 0.01$ for seed mass is not a negative correlation– is the minus sign missing here?

>>> Modified and we have revised all correlation signs in the text.

L 382 “perturbated” -> “perturbed”

>>> Modified.

Methods should be written consistently in the past tense, e.g. L 435 “select” -> “selected” ; L 510, 516, 521 “run” -> “ran”

>>> Methods have been carefully revised and corrected to the past tense.

L 547 “compare” -> “compared”

>>> Modified.

Missing articles:

L 431: should be “a few thousand years ago”

>>> we have removed this sentence from the text.

L 435: “the South relict genetic cluster...”

>>> **Modified.**

L. 452 word missing- should be “under well watered conditions”

>>> **Modified.**

L 529: “sequenced” -> sequences

>>> **Modified.**

Reviewer #3 (Remarks to the Author):

I appreciate the clarification into the classification of the accessions. However, I remain unclear about the rationale behind the groups used. The choice of north, central and south, suggest the importance of latitudinal variation, but investigation of its existence is not discussed. Most of the study is about comparing relicts to non-relicts; but the biggest significant differences are between central relicts, and north/south non-relicts. It is not clear why this should be the most interesting comparison, or the one expected. Any differences that are SOLELY due to relict vs. non-relict status, should be most directly tested by comparing south relict to non-relict. Any statistically significant difference among all 5 groups only indicates that there is variation among groups.

This study clearly shows that central non-relict accessions are quite different from south and north relicts, but given that this difference is not replicated in comparisons of relict and non-relict accessions in the north and south, I don't think is possible to assign this difference unambiguously to the fact that central accessions are non-relicts (why do the authors think that differences between relicts and non-relict change in direction in the north and south for most traits?). Hence, the request for an overall toning down of the conclusions (not only in the example from the last review).

>>> **We thank the reviewer for pushing us to better explain the classification of accessions used here. We thoroughly revised the text to better introduce the rational and main objectives of the study. In this new version, we have introduced a justification of the variation of habitat suitability and phenotypic variation for *A. thaliana* across Europe (L112-134 in the introduction and L 372-386 in the discussion). Based on previous studies, *A. thaliana* distribution seems to be constrained by climate-associated factors associated with latitude¹. Moreover, some studies demonstrated differential responses in flowering time, growth rate and resource-use traits among central and marginal genotypes in Europe, suggesting a phenotypic center-to-margins gradient²⁻⁶. One of the main reasons determining habitat suitability and promoting the phenotypic differentiation across latitude may be attributable to peculiar climatic conditions at both range margins and the partial isolation of *A. thaliana* populations in these areas. For instance, Iberian Peninsula and Italian Peninsula located at south Europe are separated by the Pyrenees mountains and Alps, respectively, and it is also the case for Scandinavian peninsula located at north Europe. Therefore, one of the main goals of our study, now rewritten to better understanding (L136-148), was to test whether traits associated with dispersal and competitive abilities exhibit a central-to-margin differentiation (i.e. center-to-north and center-to-south). In turn, this hypothesis is also closely supported by the demographic history of *A. thaliana* (following the second colonization wave of Europe by non-relicts from central**

Europe). Following the suggestions of considering clustering of accessions according to both geography and genetics, we have conducted a new analysis to perform genetic clustering of all accessions, using the ADMIXTURE software on the full genomic sequences (after LD pruning, see M&M; L459-473). The analysis revealed that the best structure for our dataset is two clusters. Interestingly, the first cluster contains all non-relict accessions, except North-Sweden lines that all clustered with relict accessions (from the South). This was surprising, since North-Sweden lines have been classified as closely related to non-relicts in previous studies^{7,8}. In contrast, our results suggest that North-Sweden lines (at least those used in this study) are actually more related to relicts than non-relicts. Moreover, this explains why we cannot solely test geographic groups, or solely test relicts vs non-relicts, because the north and south geographic groups contain accessions from very different genetic origins (genetic clusters 1 and 2), due to (recent) migrations of central genotypes to peripheral areas. Consequently, we categorized accessions into five groups at the intersection of geographic areas and genetic origins:

“Central-GeneticCluster1” = ‘Center cosmopolitan’ in the text;

“South-GeneticCluster1” = ‘South cosmopolitan’ in the text;

“North-GeneticCluster1” = ‘North cosmopolitan’ in the text;

“South-GeneticCluster2” = ‘South relict’ in the text; and

“North-GeneticCluster2” = “North relict” in the text.

We have included a justification of the importance of phenotypic variation across latitude (L58-74), we have introduced and further discussed about the habitat suitability and phenotypic diversity for *A. thaliana* across Europe (L112-134 in the introduction and L 372-386 in the discussion), we have now included a better explanation about how geographic areas were assigned and compared (L166-187 in results and L449-L473) and we have modified Fig 1A removing ellipses to avoid misunderstanding in the geographic areas and genotype selection. We have additionally reorder and slightly rewritten some paragraphs of the result section to better introduce the main ideas of our study (see result section from L153 onward).

The current editing of the manuscript has unfortunately not solved the mismatch between aims, experimental design and conclusions:

*Aim 1: to assess the spatial structure of phenotypic variation in key traits related to dispersal, competition and stress-response abilities in *A. thaliana*.* This aim is very unclear. What is the null expectation for such aim? No differences between all 5 regions? Would one significant difference be sufficient to say there is “spatial structure”? If there is one region different what would this tell us about big patterns of trade-offs?

>>> Yes, the null expectation would be no difference in traits between groups. We agree that significant but random (or stochastic) variations between groups would not be sufficient to talk about a “spatial structure”. However, the demographic history of *A. thaliana* also suggests that there could be a central-to-margin gradient of trait differentiation, what we called here a “spatial structure”, which was tested in

the present study. Alternatively, traits might covary with latitudinal climatic variation. For clarity, we rephrased the whole introduction and the aims of the study (L 50-148).

The first aim is now to test whether traits differentiate between *A. thaliana* populations according to a center-to-margin gradient, with northern lines displaying similar trait variations as their genetic relatives at the opposite southern edge.

Aim 2 : to test whether competition-colonization trade-offs determine the distribution of A. thaliana lines, The methodology of this study only allows inferences about correlational patterns, not whether the trade-off determined the distribution of *A. thaliana*. This would require a phylogenetic approach.

With regards to this aim is also worth pointing out that the R values supporting the idea of a genetic trade-off (figure 9A) are quite small (except for the negative trade off between fecundity under control and competition). All R values associated with seed mass is $<0.01!$

>>> We agree with reviewer that the term “determine” was not appropriate, we rephrased the second aim of the study into “testing whether trait variation among geographic and genetic groups are related to major ecological trade-offs between stress resistance, competition tolerance, and dispersal ability”. Regarding the values reported for the coefficients of correlation, we agree that R values at the whole-genome level (Supplementary Fig. 9A) are low (but significant for almost all pairwise correlations). However, this is expected since, at the whole-genome level, many SNPs vary due to drift or random factors, or are involved in other adaptive traits unrelated to dispersal and competitive abilities (e.g., floral traits, defense-related traits). Actually, significant correlations at the whole-genome level are quite remarkable, even with low R coefficients. Moreover, R values are much higher when looking at pairwise correlation between 1% top-SNPs (Supplementary Fig. 9B; R is above 0.3 or below -0.3 for many trait-trait relationships), suggesting that many of the SNPs involved in the variation of a given trait also impacted the variation of another trait (also supported by the number of SNPs in common between traits, Supplementary Fig. 10). However, we toned down some statements about the genetic trade-off as suggested (eg. L38, 339;436).

Aim 3: to explain the demographic success of the nonrelict lineages in Europe.

I cannot see how any of the data collected can “explain the demographic success”, but I believe what is being hypothesized here is the success of non-relicts is associated with increase in competitiveness at the cost of fecundity.

If that is the case, the aim should be rewritten to clearly state what is being tested, and the conclusion should be that this is likely, but not unambiguous.

>>> We agree that we needed to tone down our conclusions, we rephrased the text in discussion and conclusion (L436-446) accordingly. Moreover, we rephrased aim 3 into “understanding why relictual genetic variation is maintained at the opposite edges of the distribution range”. However, we respectfully disagree that “any of the data collected can explain the demographic success of the nonrelict lineages”. Our results show that non-relicts differentiated phenotypically and get more similar to relicts (probably following hybridization and selection) as they migrated toward both north and south regions. It strongly

suggests that cosmopolitans, which have traits associated with high dispersal ability (not the opposite as said by the reviewer), become less fit in peripheral environments, where the “relict phenotype”, associated with competitive ability, is more advantageous. Again, we rephrased the text in several places to clarify our conclusions (and their limits).

Other points:

Given that none of the aims is focused on “central” or “margins”, the first subtitle of the results should be changed to reflect aim 1; or aim 1 needs to be changed to reflect this first sub-title. In this section variation in phenotype between relict and non-relict populations seem to be the main focus.

>>> We agree and modified the titles accordingly and rewritten part of the text (L112-148; L152-203)

Central non-relicts response to water stress is NOT significantly different to the other groups (Figure 1), and line 183 should be modified to reflect this.

>>> We have rephrased it.

There is good evidence that seed size has been under selection (Figure 2), but very weak indication that plant height (key for dispersal, according with the manuscript) has been under selection. Interestingly, genetic differentiation for fecundity between southern relicts and central non-relicts are also not significant. Thus, again the results are very interesting, but does not unambiguously suggest that relicts were selected for dispersal.

>>> We agree and modified the text to tone down the conclusions, and discuss the different patterns between traits (e.g., L300-306; L414-423).

Sentence added on line 106 of the word document does not make sense.

>>> Modified.

Line 232, the data being discussed in here is shown where? No figures referred....

>>> We acknowledge the reviewer's concern; nevertheless, there is no specific figure illustrating the statistics 'S' derived from Ovaskainen and colleagues. The resultant values for the statistic 'S' calculated for each trait are incorporated into the main text, and the creation of a bar plot solely for the representation of the 'S' value would be redundant.

REFERENCES

1. Yim, C. *et al.* Climate biogeography of *Arabidopsis thaliana*: Linking distribution models and individual variation. *Journal of Biogeography* **00**, 1–15 (2023).

2. Li, B., Suzuki, J.-I. & Hara, T. Latitudinal variation in plant size and relative growth rate in *Arabidopsis thaliana*. *Oecologia* **115**, 293–301 (1998).
3. Estarague, A. *et al.* Into the range: a latitudinal gradient or a center-margins differentiation of ecological strategies in *Arabidopsis thaliana*? *Annals of Botany* **129**, 343–356 (2022).
4. May, R.-L., Warner, S. & Wingler, A. Classification of intra-specific variation in plant functional strategies reveals adaptation to climate. *Annals of Botany* **119**, 1343–1352 (2017).
5. Sartori, K. *et al.* Leaf economics and slow-fast adaptation across the geographic range of *Arabidopsis thaliana*. *Sci Rep* **9**, 10758 (2019).
6. Hopkins, R., Schmitt, J. & Stinchcombe, J. R. A latitudinal cline and response to vernalization in leaf angle and morphology in *Arabidopsis thaliana* (Brassicaceae). *New Phytologist* **179**, 155–164 (2008).
7. Lee, C.-R. *et al.* On the post-glacial spread of human commensal *Arabidopsis thaliana*. *Nature Communications* **8**, 14458 (2017).
8. Exposito-Alonso, M. *et al.* Genomic basis and evolutionary potential for extreme drought adaptation in *Arabidopsis thaliana*. *Nat Ecol Evol* **2**, 352–358 (2018).